# A systemic and comprehensive assessment of coastal hazard changes: method and application to France and its overseas territories

Marc Igigabel[1], Marissa Yates[2,3], Michalis Vousdoukas[4], Youssef Diab[5]

[1]Cerema, Technopôle Brest Iroise, 155 rue Pierre Bouguer, BP5, 29280 Plouzané, France
[2]LHSV & Cerema, 6 quai Watier, BP 49, 78401 Chatou, France
[3]LHSV, Ecole des Ponts, EDF R&D, 6 quai Watier, BP 49, 78401, Chatou, France
[4]European Commission Joint Research Centre, Via Enrico Fermi 2749, I-21027-Ispra, Italy.
[5]Université Gustave Eiffel, Lab'Urba, 5, Boulevard Descartes, 77430 Champs-sur-Marne, France

*Correspondence to*: Marc Igigabel (marc.igigabel@cerema.fr)

**Abstract.** In the context of climate change, height and frequency variations of extreme sea levels (ESL) are studied using deterministic and probabilistic approaches. However, this type of approach does not highlight the dynamic effects (waves, currents) generated by metocean events (storms, cyclones, long swells and tsunamis), beyond their effects on sea levels. In particular, ESL estimates are calculated by considering the main determining physical factors but cannot include all the effects of these factors. Ultimately, this can lead to confusion between ESL and hazard. This article proposes a systemic assessment method to analyze coastal hazard changes at regional scales, integrating parameters influencing sea levels, as well as factors describing the geomorphological context (length and shape of the coast, width of the continental shelf), metocean events, and the marine environment (e.g., coral reef state and sea ice extent). French mainland and overseas territories were selected to apply the method. The present study highlighted the need to consider not only the sea level variability, but also the current and future characteristics of metocean events. The long, concave coasts bordered by a wide continental shelf appear particularly sensitive to variations in the intensity or trajectory of metocean events. Coral reef degradation in the tropics and the decrease in seasonal sea ice extent in the polar regions can also significantly change the nearshore hydrodynamics and impacts on the shoreline. These results help to predict the types of hazard (shoreline erosion, rapid submersion and/or permanent flooding) that will increase the most in different coastal zones.

## 1 Introduction

In 2001, the frequency and intensity of extreme climate events, including marine flooding from tropical and other storms was identified by the Intergovernmental Panel on Climate Change (IPCC) as one of the five « reasons for concern » (IPCC, 2001, 2007). Recent research has improved knowledge of the mechanisms generating sea level rise (Cazenave and Llovel, 2010; Cazenave and Le Cozannet, 2013; Hamlington et al., 2020; Fox-Kemper et al., 2021). Frederikse et al. (2020) showed that since 1900 the sum of the contributions to sea level change is consistent with the trends and multidecadal variability in

observed sea level on both global and basin scales. However, significant uncertainties remain about the evolution of hazards at the coastline generated by the marine environment. In this article, these hazards will be referred to as « coastal hazards », with the understanding, however, that the coast may be affected by other sources of hazard (intense precipitation, overflow of streams, etc.).

The issue of the evolution of coastal hazards is often addressed through the evolution of extreme sea levels (ESLs) along the coast. For example, Vitousek et al. (2017) warned that the 10 to 20 cm of sea level rise expected no later than 2050, will more than double the frequency of extreme water-level events in the tropics. Vousdoukas et al. (2018) projected a very likely increase of the global average 100-year ESL between 2000 and 2100 of 34–76 cm under a moderate-emission-mitigation-policy scenario and of 58–172 cm under a business as usual scenario. Under these scenarios, a large part of the tropics would

be exposed annually to the present-day 100-year ESL in 2050. These assertions concern the evolution of water levels at the coast, but are not sufficient to describe the evolution of coastal hazards. Yet many authors evaluating changes in coastal risks restrict the representation of the hazard to ESLs (e.g. Hallegatte et al. (2013), Tiggeloven et al. (2020), Rasmussen et al. (2022)). This kind of analysis uses a deterministic approach based on a single factor (the ESL that is expressed by two parameters: maximum water level and frequency) to represent a complex hazard that can only be fully addressed using a

systemic approach that considers multiple factors (Igigabel et al., 2021). In comparison, on land, systemic approaches are used in studying surface runoff and defining strategies in water resource and flood risk management (Shaikh et al., 2022; Verma et al., 2023; Mehta et al., 2023).

To answer this problem in the coastal domain, it is necessary to highlight that during a metocean event, the phenomena of

flooding and coastal erosion are not only determined by the maximum sea level, but also by coastal waves and currents, and overtopping and overflow discharges over flood protection structures (Formentin et al., 2018; Igigabel et al., 2022). Estimating these discharges and hydrodynamic conditions for the duration of an event requires a good understanding of the physical phenomena that generate the hazard. Retrospective analyses of events help to understand correctly the mechanisms that cause the observed flooding or erosion. For example, by simulating Total Water Levels (TWLs) along the Bight

extending from North Carolina to Florida during three historical Tropical Cyclones (TCs) with similar tracks, Hsu et al. (2023) found that the magnitude and duration of the increase in TWLs and wind waves are influenced by TC intensity,

translational speed and distance from the shore. In particular, the authors showed that a decrease in TC translation speed led to longer exceedance durations of TWLs, which may result in larger impacts. However, it is not possible to predict deterministically the physical characteristics of future events, nor to assess the corresponding hydrodynamic conditions. To compensate for this, probabilistic approaches have been developed. For example, using a large number of synthetic hurricanes that consider the natural variability in hurricane frequency, size, intensity and track, Krien et al. (2015) estimate the 100-year and 1000-year surge levels for the archipelago of Guadeloupe. Following the same principle, Krien et al. (2017) estimate 100-year surge levels in Martinique for the present climate or considering a potential sea level rise. These results help to determine the necessary levels of protection structures in the short and medium term. However, a single parameter (the maximum water level) is not enough to characterize the hazard and define all the crisis management measures, particularly when water levels exceed the level of protection or when protection structures fail (for example, breaches in levees or dunes). In addition, the accuracy of these estimates decreases in the long term due to: (i) high uncertainties in sea levels beyond 2050 (IPCC, 2019); (ii) the increase in the proportion of high-intensity cyclones worldwide (Masson-Delmotte et al., 2021); and (iii) environmental instability, notably because of geomorphological (e.g., subsidence, coastline retreat) and biological (e.g., degradation of coral reefs and mangroves) changes. These changes modify hydrodynamic and hydrosedimentary processes on the coast both in the long term and during individual events.

With the aim of making progress in the global assessment of coastal hazards in the long term, the guiding principle of this paper is to promote the use of the latest advances in research on changes in metocean events and water levels, while also showing the importance of studying other factors whose evolution is more predictable than storm surge and that may be equally important, namely: tidal regimes, geomorphological settings and environmental changes (particularly those modifying hydrodynamic conditions at the coast).

Although water levels are not the only parameter to consider, it is necessary to begin by clarifying the definitions of the different levels to which we will refer regularly.

First, it is necessary to distinguish between global mean sea level (GMSL) and relative sea level (RSL). The GMSL rise can be defined as the global change in mean sea level (MSL) relative to the terrestrial reference frame, due to the combined effects of change in the volume of the ocean and change in the level of the sea floor (Church et al., 2013). The RSL rise can be defined as the change in sea level with respect to the land (Lowe et al., 2010).

Second, it is necessary to assess the main factors influencing the evolution of water levels, whether they are GMSL, RSL or ESLs.

MSL changes, both globally and locally, vary according to seasonal, annual, and longer time scales. These variations may be caused by changes in the mass of water in the ocean (e.g., due to melting of glaciers and ice sheets and changes in terrestrial water storage), by changes in ocean water density (e.g., volume expansion as the water warms), by changes in the shape of the ocean basins (e.g., due to plate tectonics) and changes in the Earth's gravitational and rotational fields, and by local subsidence or uplift of the land (Oppenheimer et al., 2019). In the future, changes in the GMSL are expected to be strongly controlled by the greenhouse gas (GHG) emission scenarios (Oppenheimer et al., 2019; Fox-Kemper et al., 2021) and cryosphere evolution: in case of faster melting of the Greenland and Antarctic ice caps, Bamber et al. (2019) estimate the increase in GMSL above 2 m in the 21st century.

The RSL will depend on both the GMSL and local vertical land movement (VLM). VLM can have natural causes (e.g. isostasy, elastic flexure of the lithosphere, earthquakes and volcanoes, landslides and sedimentation) and, more locally, anthropogenic causes, in particular, soil loading, extraction of hydrocarbons and/or groundwater, drainage, mining activities (Gregory et al., 2019). During the 20th century, the coasts of Tokyo, Shanghai, and Bangkok collapsed by several meters as a result of water extraction and soil loading from the construction of buildings and infrastructure (Nicholls et al., 2008). Subsidence rates of the order of 1 to 10 cm/year are commonly measured in coastal megacities and are exceeding sea level change rates by a factor of ten (Erkens et al., 2015). However, even in the absence of subsidence, the majority of coastlines are expected to experience RSL changes on the order of 30 percent of GMSL rise (Gregory et al., 2019; IPCC, 2019; Dayan et al., 2021).

ESLs are also projected to change due to changes in RSL, tides, wind waves, and storm surges (Vousdoukas et al., 2018). Storm surges are defined by Gregory et al. (2019) as « the elevation or depression of the sea surface with respect to the predicted tide during a storm ». The predominant factor in the evolution of ESLs is the variability in sea levels generated by astronomical tides and storm surges (Buchanan et al., 2016). The combination of tide and storm surge phenomena requires a statistical approach. Thus, the change in ESL events is commonly expressed in terms of an amplification factor and an allowance. The amplification factor denotes the amplification in the average occurrence frequency of a certain extreme event, often referenced to the water level with a 100-year return period estimated from historical data. The allowance denotes the increased height of the water level [m] with a given return period. This allowance equals the regional projection of RSL rise with an additional height related to the uncertainty in the projection (Hunter, 2012). Amplification factors are strongly determined by the local variability in ESL events. Locations where this variability is large due to large storm surges and astronomical tides will experience a moderate amplification of the occurrence frequency of extremes in comparison to locations with small variability. Globally, this contrast between regions with large and small amplification factors becomes clear for projections by mid-century and considerable in the coming centuries (Vitousek et al., 2017). In particular, many coastal areas in the lower latitudes may expect amplification factors of 100 or larger by mid-century, regardless of the GHG

emissions scenario. By the end of the century, and in particular if GHG emissions are not reduced, such amplification factors may be widespread along global coastlines (Vousdoukas et al., 2018).


The objective of this paper is to present a comprehensive method to assess the evolution of coastal hazards at regional scales in the context of climate change. The proposed systemic method emphasizes the need to focus on the analysis and interpretation of the modelling results, by putting them into perspective with respect to the biophysical conditions (both current and forecasted).


This method was developed as the result of an empirical process, considering diverse situations. This process has the advantage of demonstrating the wide applicability of the proposed method. The case studies used to develop the method are in France and French territories, but they are located in different latitudes (equator, tropics and temperate zones) where they are exposed to different climates and are characterized by different geomorphological configurations (including continental

or island). This study highlights that considering the qualitative factors describing the geomorphological context, metocean events and the marine environment in a systemic approach was necessary to assess the evolution of hazards.

To ensure that the method can be applied for operational purposes, the use of freely accessible data has been promoted. However, applications of this method should include a thorough bibliographic analyse or even additional observational or

modeling studies at the chosen case study sites. It is important to emphasize here that the application of the method depends strongly on the available data, and therefore it is necessary to gather the best data for each site. Since the quantity and quality of data is not the same everywhere, the uncertainties in the results will also vary and must be addressed.

Finally, following the application of the method to the case studies, the results obtained using this systemic approach can

contribute to improving predictions of the evolution of different types of coastal hazards (shoreline erosion, rapid submersion and/or permanent flooding).

## 2 Method for assessing changes in coastal hazards at regional scales

To assess the evolution of coastal hazards, at regional and multidecadal scales, it is necessary to combine a quantitative approach evaluating the parameters characterizing ESLs and a qualitative approach analyzing other factors more difficult to

quantify, such as the metocean event type, the geomorphological configuration of the coast and other environmental factors (e.g. coral reef degradation in the tropics and decrease in seasonal sea ice extent in the polar regions) that may change the hydraulic actions on the shoreline.

Figure 1 shows the three proposed steps. These three stages will be presented in succession. In applying the method, however, it is possible to implement the first two steps in parallel.

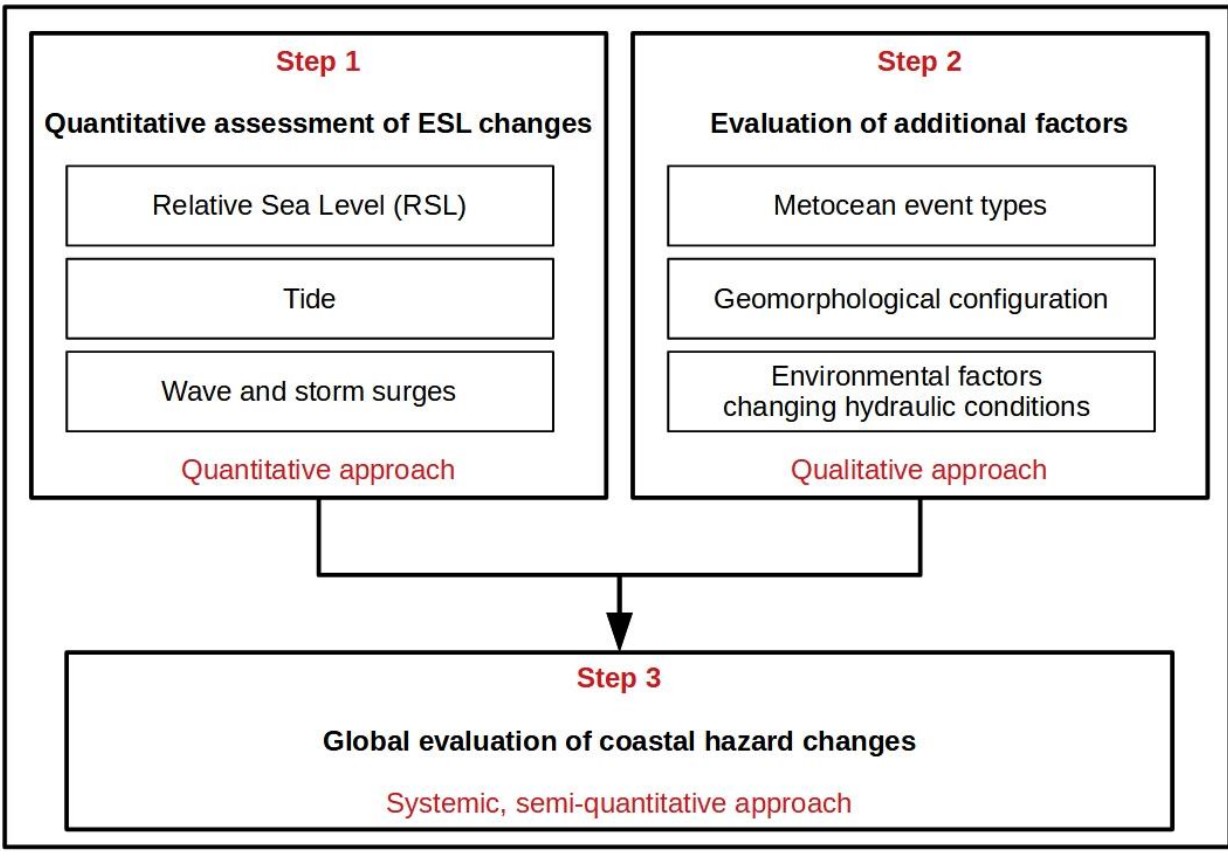


**Figure 1: Method for assessing changes in coastal hazards**

Before presenting these three steps, it is essential to remember that flood risk, and more generally, all coastal risks, can be defined in at least two alternative ways (FLOODsite, 2009). The first definition considers risk to be the result of the exposure of a vulnerable stake to a hazard, which is reflected in the following formula:

risk = hazard (metocean event) * exposure * vulnerability (of the society/area/structure)                    (1)

However, in an attempt to quantify risk, and considering that the word « risk » suggests a probability of occurrence, a second definition may be highlighted:

- the two terms « exposure » and « vulnerability » can be substituted by « consequences », with the consequences
being generally more quantifiable (for example, in number of fatalities and economic damage) than the previous two terms;

- the hazard can be represented by its probability distribution.

This yields the second definition:

risk = probability (of the hazard) * consequences                                     (2)

This article is mainly devoted to the evaluation of the hazard. Consequences (e.g. erosion and flooding) will only be addressed in the discussion. However, reference to risk definitions is necessary because the expression of hazard depends on the definition of risk.


Quantitative assessments of ESL changes (Step 1 of Fig. 1) are probabilistic and therefore fall within the second definition of risk. The challenge of the proposed method is to use these evaluations, in combination with other factors, such as the types of metocean events, the geomorphological configuration of the coast and changes in the marine environment affecting hydraulic conditions. These « additional factors », which are more difficult to quantify, will be studied in a second, parallel

phase (step 2 of Fig. 1). The last step is to describe the evolution of the hazard within the framework of a systemic approach. In this step (step 3 of Fig. 1), the first definition of risk is applied to search for a sufficiently synthetic way of presenting the data to produce an overall assessment.

Given the multiple effects of climate change on sea levels (GMSL and RSL), atmospheric conditions, and wave conditions,

the evolution of the factors and parameters can only be assessed using global development hypotheses. To represent different trajectories, the IPCC identified five scenarios referred to as SSPx-y, where 'SSPx' refers to the Shared Socio-economic Pathway ('SSP' describing the socio-economic trends underlying the scenario), and 'y' refers to the approximate level of radiative forcing (in W $m^{-2}$) resulting from the scenario in the year 2100. To determine the most relevant scenario and identify the most appropriate time horizon, an exposure and vulnerability analysis and an assessment of the duration of the

project are required. These reflections are an essential prerequisite for the application of the proposed method.

For the assessment of sea levels, Table 1 shows the projections for the GMSL under the SSP5-8.5 scenario at three time scales: 2050, 2100 and 2300.

**Table 1: GMSL projections for 2050, 2100 and 2300 for the SSP5-8.5 scenario. Median values and ranges (17th to 83rd percentiles) are shown using the 1986-2005 period as a reference (IPCC, 2019).**

| Climate scenario | 2050 Median | 2050 17-83% range | 2100 Median | 2100 17-83% range | 2300 Median | 2300 17-83% range |
|---|---|---|---|---|---|---|
| SSP5-8.5 | + 32 cm | 23 cm to 40 cm | + 84 cm | 61 cm to 110 cm | +385 cm | 230 cm to 540 cm |

The GMSL values displayed in Table 1 show high uncertainties in the long term. For example, in 2100, the median value is 0.84 m and the 17th to 83rd percentile range is 0.61 to 1.10 m (the high value is almost double the low value). These GMSL

estimates were established for the SSP5-8.5 scenario, but greater differences exist if other scenarios are considered in a complementary manner. Similar (or higher, since other local phenomena must be considered) uncertainties exist for the RSL and the centennial ESL.

In conclusion, to make long-term management and adaptation décisions, a general description of the evolution of the hazard, including the estimated uncertainties, is considered more appropriate than a forecast of the evolution of a parameter at a given date, which is likely to be imprecise. Accordingly, conservative assumptions will be used when assessing sea level components in this method.

### 2.1 Step 1: ESL change assessment

Tidal simulations show no significant impacts of RSL rise on tidal amplitude during the 21st century at regional scales, although this does not exclude potential local effects (Pickering et al., 2012; Haigh et al., 2020). For example, Idier et al. (2017) showed that notable increases in high tide levels occur in the northern Irish Sea, the southern part of the North Sea and the German Bight, and decreases occur mainly in the western English Channel. Depending on the location, they can account for $+/-15\%$ of RSL rise (as long as RSL rise remains smaller than 2 m). Given the strong uncertainties of RSL rise, it is assumed in this study that the tides and tidal range remain constant.

ESLs are the result of the combination of the RSL, the tide level, and the storm surges produced by metocean events (Calafat et al., 2022). To understand the evolution of ESLs, it is assumed that metocean events simultaneously generate atmospheric storm surge (caused by atmospheric depressions, wind) and energetic waves. At the coast, waves also contribute to the total water level by two phenomena:

- wave set-up: a time-varying (on the order of 30 minutes), wave-driven increase in the mean water level at the coast, resulting from wave dissipation in the nearshore (Bowen et al., 1968);
- run-up: the time-varying (on the order of the wave period, i.e. a few second, or the period of infragravity-waves, i.e. a few minutes) water level reached by a wave on a beach or a coastal structure, relative to the static water level, measured vertically (run-up may generate overtopping which, unlike overflowing, occurs intermittently).

In the proposed method, storm surge and wave set-up will be considered to estimate the ESL, but wave run-up will be neglected because of the challenge associated with modeling it at fine scales (Aucan et al., 2018; Gomes da Silva et al., 2020).

To fully account for changes in ESLs, the analysis must include both changes in the average occurrence frequency of a certain extreme event and the increased water level (in absolute terms and as a percentage) with a given return period. For this reason, the second definition of risk will be used to study the joint probabilities of the parameters impacting the sea

level, namely the RSL, the tidal range and the storm surge (including the effects of waves). Following Vousdoukas et al. (2017) changes in the magnitude and frequency of occurrence of the present 100-year ESL (ESL$_{100}$) are evaluated. It is assumed that ESL are driven by the combined effect of MSL, tides ($\eta_{tide}$) and water level fluctuations due to waves and storm surges ($\eta_{w-ss}$). As a result, ESL can be defined as:

$$ESL = MSL + \eta_{tide} + \eta_{w-ss} \tag{3}$$

We must here appreciate the need to use a more detailed method for each particular case, because according to Idier et al. (2019), depending on the type of environments (e.g., morphology, hydrometeorological context), non-linear interactions between various components - tide, surge, wave set-up - can reach several tens of centimeters. For instance, using a numerical modeling approach, focusing on the German Bight area (SE of North Sea) and assuming a sea level rise of 0.54 m, Arns et al. (2015) show that taking into account the interactions between mean sea level, tide and atmospheric surge leads to positive changes in extreme water levels relative to the MSL rise. The largest non-linear increases in the order of +0.15 m occur in the shallow areas of the Wadden Sea. However, introducing a novel approach to statistically assess the non-linear interaction of tide and non-tidal residual, Arns et al. (2020) demonstrate that extreme sea levels are up to 30% (or 70 cm) higher if non-linear interactions are not accounted for. The largest effects of Tide Surge Interaction (TSI) are found for the US East Coast and the Gulf of Mexico, the UK North Sea coastline, and parts of the southern Japanese coast. The highest value, 66 cm, is found in Cromer along the UK North Sea coastline. On the selected coasts, other values are generally between 20 and 50 cm. In conclusion, while statistical methods demonstrate that, globally, non-linear interaction modulates extreme sea levels, numerical modeling shows that non-linear interaction can locally induce increases up to tens of centimeters. Given the high uncertainties regarding the long-term evolution of sea levels and geomorphological changes, we shall consider that equation (3) provides in the general case a conservative estimate of the ESL and, if not, that the other sources of uncertainty are greater than the water level rise related to these non-linear interactions.

The climate extremes contribution $\eta_{w-ss}$ from waves and storm surge can be estimated as:

$$\eta_{w-ss} = SSL + 0.2\, H_s \tag{4}$$

where SSL is the storm surge level, $H_s$ is the significant wave height and $0.2\, H_s$ is a generic approximation of the wave set-up (U.S. Army Corps of Engineers, 2002). This equation is a conservative estimate of the wave set-up, which mostly depends on the local slope, breaking wave height, and wave period (Stockdon et al., 2006), and may be closer to 10% of the breaking wave height. Larger values may only be observed at steep sandy beaches (e.g. Martins et al., 2022) or steep shore platforms (Sheremet et al., 2014; Lavaud et al., 2022). In the current state of the art, numerical simulations of set-up have been carried out only locally (e.g., Lange et al. (2021), van Ormondt et al. (2021)). Such retrospective simulations are computationally expensive (e.g. to simulate accurately set-up, model grid resolution of ~10-50 m may be needed for study sites with large variations in local bathymetry and the wave field). In addition : (i) accurate and high resolution wave set-up modeling requires high resolution bathymetric data (Stephens et al., 2011), which are not available at the spatial scales of our case study; (ii) significant morphological changes are expected in nearshore areas, especially with rising sea levels, which

can have a significant effect on the wave set-up at different timescales (e.g. especially for sandy beaches exposed to waves: Ruggiero et al., 2001 ; Thiébot et al., 2012 ; Brivois et al., 2012). Thus, at large spatial (e.g. global) and long temporal (e.g. 21st century) scales, simplified models such as the one used here (Vousdoukas et al., 2018) may provide a first estimate of the expected wave set-up.

## 2.2 Step 2: Evaluation of additional factors

Two types of regional factors can be considered essential for all coastlines: phenomena likely to generate energetic sea states and the geomorphological configuration. In addition, other environmental factors affecting hydraulic conditions should be considered in the tropics (in particular, coral reef state) and at the poles (in particular, sea ice extent).

### 2.2.1 Metocean event types

Three main coast categories are considered in this method:  exposure to no particular phenomenon, to storms, or to cyclones (which also implies exposure to storms). On these three categories are superimposed (i) the potential exposure to long swells generated by a metocean event far from the studied coastline (ii) the potential exposure to tsunamis, since the origin of tsunamis is not related to the meteorological conditions.

The increase in wave damage could also be assessed, considering local changes in significant wave height (average height of the highest one-third of the waves in a given sea state). However, trends in coastal wave climates are reported with a low confidence level in the IPCC (2019) report. Therefore, these trends will not be explored in detail: the focus will be on the strong differences that already exist between the wave climates of the maritime facades.

### 2.2.2 Geomorphological configurations of the coast

It is necessary to identify characteristics of the coast that influence the different components of the surges.

Storm surges are controlled not only by the characteristics of metocean events (e.g. for tropical cyclones (TC), the TC intensity, the distance to the TC eye, the TC heading direction and the TC translation (Hsu et al., 2023)), but also by the width of the continental shelf and the relative water depth (Kennedy et al., 2012). For example, during the 20-year study period (1998-2018), storm surges hardly reached 1.0 m along the coastlines of the southern Bay of Biscay and the eastern Mediterranean Sea, but exceeded 2.0 m in the English Channel (Dodet et al., 2019). In shallow water, the wind effects can significantly dominate the effect of the atmospheric pressure, and the surge can be strongly amplified by the bathymetry along the coast (Bertin et al., 2012). The width and depth of the continental shelf have an important influence on the magnitude of storm surges.

Secondly, in regions exposed to cyclones, the surges measured on islands are much lower than those measured on continental facades. For example, on the islands of the West Indies, the storm surges rarely exceed 3 m (Krien et al., 2015 and Krien et al., 2017), whereas in the case of Katrina, which impacted the United States in 2005, surges in eastern Louisiana reached values between 3.05 m and 5.79 m (Graumann, 2006). During the same event, the surges exceeded 8 m at several locations along the Mississippi coast (Dietrich et al., 2010). This difference between the surges observed on continental coasts and island coasts is explained by the fact that, in the case of a hurricane, the impact of the low pressure associated with the storm on surge is minimal in comparison to the water being forced toward the shore by the wind. However, in the case of small islands (e. g. West Indies or La Réunion), the surge is generally reduced: wind-forced water accumulates less on short shorelines than on long shorelines (Durand, 1996). Therefore, the length of the maritime facade appears to be the second determining factor of the surges.

Thirdly, the maximum potential storm surge for a particular location is sensitive to small changes in storm intensity, forward speed, size, angle of approach to the coast, central pressure, and also the shape and characteristics of coastal features such as bays and estuaries (Flather, 2001; Rego and Li, 2010 and Kennedy et al., 2012). Wind-forced water will accumulate less on convex-shaped shorelines than on concave-shaped shorelines (Bertin et al., 2012; Krien et al., 2015; Krien et al., 2017). Therefore, the shape of the coast appears to be the third determining factor of the surges.

Consequently, in order to account for the main geomorphological parameters determining the magnitude of the surge, we propose to use three criteria:

- the length of the coast;
- the width of the continental shelf;
- the convex or concave shape of the coastline.

### 2.2.3 Other environmental factors changing hydraulic conditions

Other phenomena induced by climate change can increase the intensity and frequency of extreme events, or increase their effects:

- ocean acidification combines with ocean warming and deoxygenation to impact ecosystems (e. g., coral reefs and oyster beds), which can reduce coastal flood protection. Under conditions expected in the 21st century, global warming and ocean acidification may compromise carbonate accretion, with corals becoming increasingly rare on reef systems. The result should be less diverse reef communities and carbonate reef structures that fail to be maintained. Climate change also exacerbates local stresses from declining water quality and overexploitation of key species, driving reefs increasingly toward the tipping point for functional collapse (Hoegh-Guldberg et al., 2007; Albright et al., 2018);

- in polar regions, the decrease in seasonal sea ice extent in the Arctic, together with a longer open water season, provide less protection from storm impacts (Forbes, 2011). In addition, a longer fetch allows more energetic waves to form, which can have consequences outside of the polar regions, for example on coasts exposed to southern swells.

These two factors that affect wave formation and propagation, as well as currents, are also considered in the proposed method.

## 2.3 Step 3: global hazard assessment

Following step 1 and step 2, the overall assessment of the hazard should be produced using the following factors and parameters (evaluated on the basis of the SSP5-8.5 scenario):

- RSL height change ($\Delta$RSL);
- tidal range;
- surge with a 100-year return period;
- ESL height variation ($\Delta$ESL);
- ESL percentage change (%$\Delta$ESL);
- return period of the present day 100-year ESL;
- geomorphological configuration of the coast;
- metocean event type; and
- other environmental factors affecting the hydrodynamics at the shoreline.

In this method, each factor should not be considered independently. On the contrary, the joint effects of the different factors should be assessed to understand the dynamics of the system. However, the diversity of the different observed contexts prevents proposing a quantitative calibration and weighting system. This is the main reason why this method is called « systemic », in reference to the study of a system (the coast). However, this method should not be called « systematic ». Even though the analysis framework incorporates multiple factors, the expert in charge of the study of a particular site should consider the qualitative and quantitative factors together. In summary, this method aims to help experts to make a structured evaluation.

### 3 Application of the proposed method to French coasts

The proposed method is applied to France and its overseas coasts (cf. Fig. 2). In the following, the islands of Saint-Martin,
Saint-Barthélemy, Guadeloupe and Martinique may be referred to as the « West Indies ».

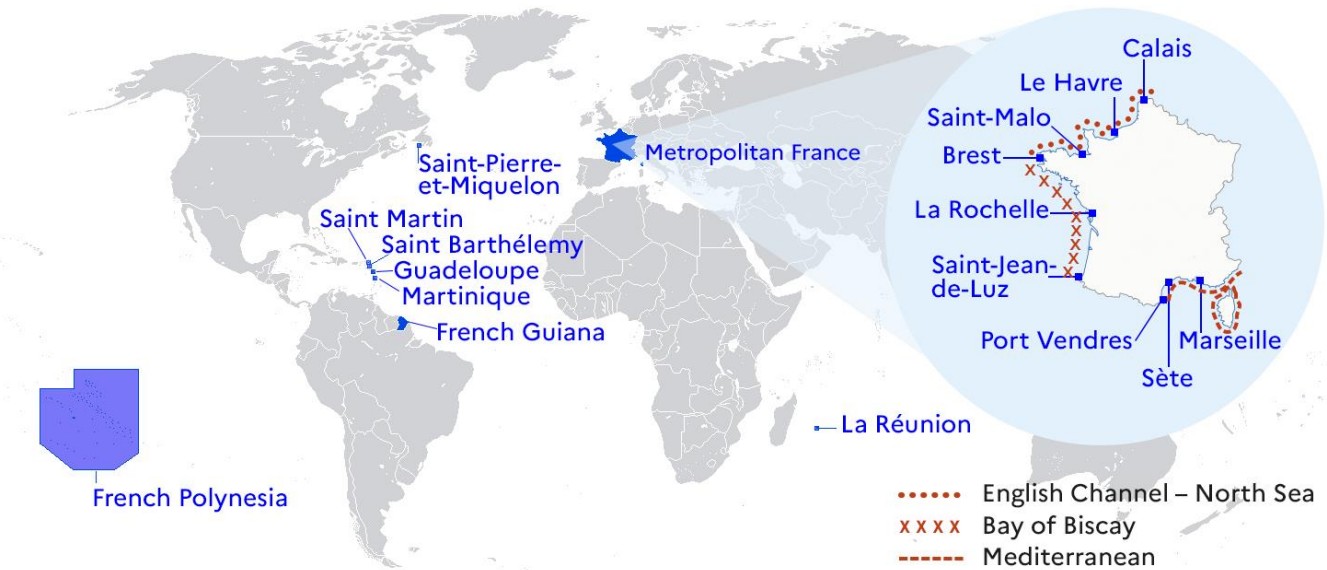

**Figure 2: A map showing the French territories considered in this study.**

The choice of the scenario and the time scales should consider the existence of the high human and economical issues at the
studied coasts (coastal cities, port and industrial facilities) and the strong uncertainties about the contributions of ice caps to
the rise in water levels (Bamber et al., 2019; Dayan et al., 2021). For these two reasons, we use the SSP5-8.5 scenario.

#### 3.1 ESL change assessment

The evolution of ESLs is assessed using the three components (Eq. 3): (1) RSL, (2) tidal range and (3) combined storm surge
and wave setup. The framework developed by Vousdoukas et al. (2018) is used to evaluate the contributions of each of these
components, of which the baseline values are calculated in global reanalyses of waves and storm surges. Then, CMIP5
models are used to estimate future relative changes to the meteorological water levels. Lastly, changes in sea level, the
astronomical tide, and meteorological water levels are combined to produce the ESL values.

#### 3.1.1 RSL: projections

Table 2 shows the median values of the RSL rise projections for the period 1995-2014 at selected points along the French
coastline. For the SSP5-8.5 scenario, the median values of the RSL rise on the French coasts are fairly uniform: between
0.16 m and 0.20 m in 2050, and between 0.76 m and 0.92 m in 2100.

**Table 2: Projections of RSL rise at selected points along the French coastline compared to the period 1995–2014 for the SSP5-8.5 scenario**

| | Geographic coordinates | | Projection of RSL rise (m) | |
|---|---|---|---|---|
| | Latitude | Longitude | 2050 | 2100 |
| Calais | 50.972122801049395 | 1.8400588679271384 | 0.19 | 0.86 |
| Le Havre | 49.485774012966544 | 0.0897840202510471 | 0.19 | 0.87 |
| Saint-Malo | 48.641170797005124 | -2.0313888026241402 | 0.19 | 0.87 |
| Brest | 48.368792381060274 | -4.4887286031866935 | 0.19 | 0.83 |
| La Rochelle | 46.14879125637811 | -1.1691588857635562 | 0.16 | 0.76 |
| Saint-Jean-de-Luz | 43.39842588942395 | -1.676715829943722 | 0.17 | 0.79 |
| Port Vendres | 42.52411089711506, | 3.1143835886292806 | 0.16 | 0.76 |
| Sète | 43.39330679508365 | 3.699492031443802 | 0.16 | 0.76 |
| Marseille | 43.29569227471328 | 5.352448215467127 | 0.17 | 0.78 |
| Saint-Pierre (Saint-Pierre-et-Miquelon) | 46.786272435631275 | -56.16190646722868 | 0.19 | 0.83 |
| Pointe-à-Pitre (Guadeloupe) | 16.23300045952869 | -61.53571250198115 | 0.20 | 0.90 |
| Cayenne (French Guiana) | 4.93572687841612 | -52.340676954198194 | 0.20 | 0.90 |
| Pointe des Galets (La Réunion) | -20.936178918145654 | 55.280686667086655 | 0.19 | 0.92 |
| Papeete (French Polynesia) | -17.53479287238126 | -149.58674796473545 | 0.20 | 0.91 |


### 3.1.2 Tidal range

Using the standard tidal classification system (Masselink and Short, 1993), the French coastline can be divided into three categories based on the tidal range data provided by SHOM (source: Data.shom.fr):

- microtidal coastline (tidal range < 2 m): the coasts of the Mediterranean with a tidal amplitude between 0.2 and 0.5
m and the coasts of the West Indies, La Réunion and French Polynesia where the tidal amplitudes are less than 1 m; in addition, the coasts of Saint-Pierre-et-Miquelon, where the amplitude reaches 1.7 m;
- mesotidal coastline (tidal range between 2 and 4 m): the coasts of French Guiana (the maximum amplitude reaches 2.9 m);
- macrotidal coastline (tidal range > 4 m): on the coasts of the Atlantic, English Channel and North Sea, with
significant differences shown in Fig. 3.

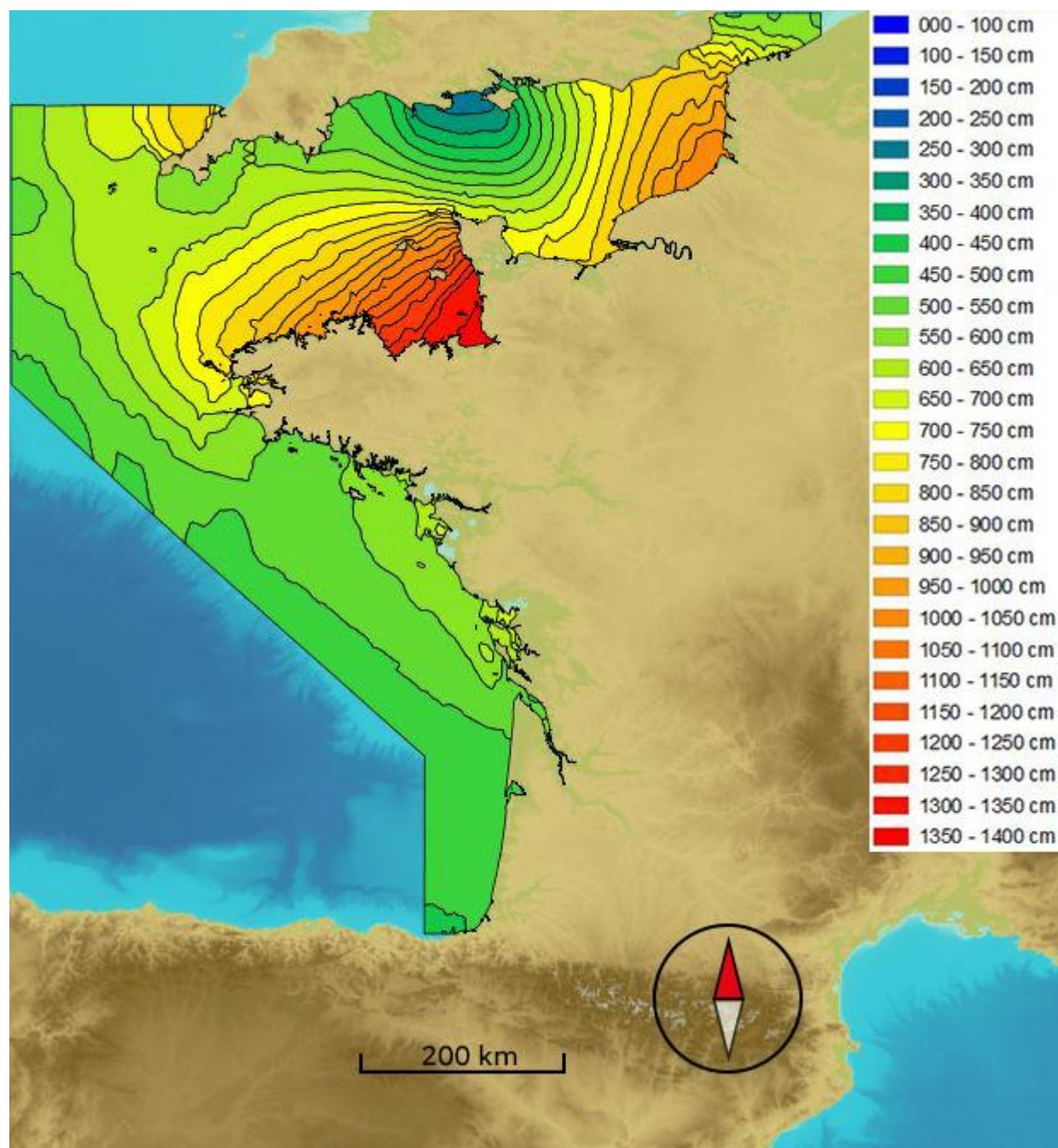

**Figure 3: Maximum tidal range (source: Data.shom.fr)**

### 3.1.3 Waves and storm surges

Information on surges will be presented for France and its overseas territories.

In Metropolitan France, there are significant variations along the coastline, as shown by the estimates of the 100-year return period storm surge presented in Table 3.

**Table 3: Estimate of the 100-year return period storm surge, calculated as the average value of statistical adjustments of a Pareto distribution (GPD) and an exponential law (Cerema, 2018) applied to storm peaks. The values of Calais, Le Havre and La**
**Rochelle have been corrected to consider historical events and the comparative study of frequency analysis approaches for extreme storm surges (Hamdi et al., 2014).**

| Tide gauge | 100-year return period surge (m) |
|---|---|
| Calais | 1.6 |
| Le Havre | 1.6 |
| Saint-Malo | 1.1 |
| Brest | 1.0 |
| La Rochelle | 1.7 |
| Saint-Jean-de-Luz | 0.6 |
| Port-Vendres | 0.9 |
| Sète | 1.1 |
| Marseille | 1.3 |

Although the relative differences are significant, they remain small in absolute terms. As mentioned in the presentation of the method, the large width of the continental shelf in the North Channel and the positioning at the center of the gulf or bay (e.g.,
Le Havre, Saint Malo, La Rochelle, Sète and Marseille) or in a strait (e.g., Calais) explain the higher values observed at these locations.

Overseas, the Cerema (2019, 2020, 2021) provides information on the storm surge measured, observed and modelled in La Réunion, French Guiana, Martinique, Guadeloupe, Saint-Martin and Saint-Barthélemy and Saint-Pierre-et-Miquelon. Surges
measured by tide gauges commonly reach values of 0.5 to 1 m. Nevertheless, these measurements are often taken in sheltered areas (e.g., ports). Additional increases in the water level along the open coast may be more or less significant depending on the geomorphological configuration. For example, during the passage of Cyclone Irma on September 6, 2017, an instantaneous surge of 2.0 m was measured at the Saint-Martin tide gauge[1]. The surge, modeled by Météo-France, was more than 3 m on the northern coasts of Saint-Martin (Marigot Bay, Grand Case) and Gustavia (Saint-Barthélemy), but
hardly more than 1.2 m on the island's almost straight coastline (De la Torre, 2017). Similarly, in September 1989, during the passage of Cyclone Hugo on the Guadeloupe archipelago, there was evidence that the sea level had increased by 2 to 3 m along the coast (Pagney, 1991; Saffache et al., 2003; Krien et al., 2015).

---

[1] https://data.shom.fr/donnees/refmar/SAINT_MARTIN

Finally, on the islands of La Reunion and French Polynesia, in addition to cyclones, southern swells represent one of the main hazards (Lecacheux et al., 2012). Continuous water height and wave measurements from the French national observation services ReefTEMPS (https://www.reeftemps.science/) and DYNALIT (https://www.dynalit.fr/) show that on coastlines directly exposed to swell (e.g. the Hermitage in La Réunion), 10-year return period events generate, at the breaking point, waves with Hs ~7 to 8 m, which induce a 1.2 m setup. Infragravity waves with Hs ~1 m can be superimposed on this set-up (Bertin et al., 2020). These values are lower in ports (e.g. Saint Gilles).

These observations confirm the need, as a complement to the measurement and modeling of surges, to study the geomorphological configuration of the coastline in order to understand the very significant differences that may appear.

### 3.1.4 ESL projection (intensity and frequency)

The framework developed by Vousdoukas et al. (2018) provides estimates of the ESL allowance for France and its overseas territories (Table 4). The results obtained in 2100 for the SSP5-8.5 scenario shows that the ESL increase in absolute value is relatively homogeneous on the various maritime facades of metropolitan France: 0.89 to 1.00 m for the English Channel, 0.74 to 0.77 m for the Bay of Biscay and 0.75 to 0.78 m for the western Mediterranean. However, the percentage variations are very different, ranging between: 16 and 22% for the English Channel, 20 and 24% for the Bay of Biscay and 52 and 63% for the western Mediterranean, in 2100. The increase will therefore be much more noticeable in the Mediterranean.

Along overseas coasts, the estimated ESL increase (absolute value) is relatively homogeneous, ranging between 0.16 and 0.24 m in 2050 and 0.85 and 0.95 m in 2100. The percent change in ESL is expected to be in the range of 8% to 18% by 2050, which is likely to be noticeable in terms of hazard intensity, particularly in Guadeloupe and French Polynesia, where the relative increases are estimated to be the largest. By 2100, the increases are projected to be around 50 to 75% (with the exception of Saint-Pierre-et-Miquelon).

Table 4: Summary of the projected absolute and relative changes of the 100-year event ESL (ΔESL and %ΔESL) for the SSP5-8.5 scenario, during the years 2050 and 2100.

| | SSP5-8.5 - 2050 | | SSP5-8.5 - 2100 | |
|---|---|---|---|---|
| | ΔESL (m) | %Δ ESL | ΔESL (m) | %Δ ESL |
| Calais | 0.21 | 4.3 | 0.94 | 19.2 |
| Le Havre | 0.23 | 4.4 | 1.00 | 19.6 |
| Saint-Malo | 0.21 | 3.8 | 0.89 | 16.0 |
| Brest | 0.22 | 5.0 | 0.96 | 21.5 |
| La Rochelle | 0.16 | 4.3 | 0.77 | 19.9 |
| Saint-Jean-de-Luz | 0.16 | 5.3 | 0.74 | 24.1 |
| Port Vendres | 0.16 | 13.3 | 0.75 | 60.7 |
| Sète | 0.18 | 12.5 | 0.76 | 52.5 |
| Marseille | 0.18 | 14.6 | 0.78 | 63.2 |
| Saint-Pierre (Saint-Pierre-et-Miquelon) | 0.24 | 8.4 | 0.92 | 31.8 |
| Pointe-à-Pitre (Guadeloupe) | 0.22 | 14.1 | 0.91 | 58.1 |
| Cayenne (French Guiana) | 0.18 | 10.6 | 0.85 | 49.2 |
| Pointe des Galets (La Réunion) | 0.16 | 9.3 | 0.88 | 52.5 |
| Papeete (French Polynesia) | 0.23 | 17.8 | 0.95 | 74.7 |

The analysis of ESL height changes should be supplemented by a frequency analysis based on Table 5. The forecasts show that sea levels with a 100-year return period could occur on an annual basis by the end of this century in mainland France (except in the center of the Atlantic facade, around La Rochelle). Some regions are projected to experience an even higher increase in the frequency of occurrence of ESL, most notably along the Mediterranean, where the present day 100-year ESL is projected to occur about ten times a year. The higher increase in the Mediterranean is closely related to the low variability of sea levels on these microtidal coastlines (Vitousek et al., 2017).

For the overseas coasts, in 2100, the present day 100-year ESL is projected to occur more frequently, approximately ten times a year, except for Saint-Pierre-et-Miquelon where this frequency would be about three times a year. The increase in frequency is likely to be noticeable in 2050, especially in French Guiana, French Polynesia and Saint-Pierre-et-Miquelon, where the return periods of the present day 100-year ESL are estimated to be only 7, 16 and 18 years, respectively.

**Table 5: Return period (in years) of the present day 100-year ESL for the SSP5-8.5 scenario in the years 2050 and 2100.**

| | Projected return period of the current 100-year ESL in: | |
|---|---|---|
| | 2050 | 2100 |
| Calais | 22.81 | 0.75 |
| Le Havre | 26.56 | 0.87 |
| Saint-Malo | 27.69 | 0.81 |
| Brest | 20.61 | 0.73 |
| La Rochelle | 36.16 | 2.59 |
| Saint-Jean-de-Luz | 33.87 | 0.63 |
| Port Vendres | 27.89 | 0.10 |
| Sète | 30.02 | 0.56 |
| Marseille | 26.88 | 0.10 |
| Saint-Pierre (Saint-Pierre-et-Miquelon) | 17.69 | 0.31 |
| Pointe-à-Pitre (Guadeloupe) | 35.14 | 0.10 |
| Cayenne (French Guiana) | 7.54 | 0.10 |
| Pointe des Galets (La Réunion) | 57.53 | 0.10 |
| Papeete (French Polynesia) | 15.86 | 0.10 |


## 3.2 Evaluation of additional factors

### 3.2.1 Metocean event types

For metropolitan France, the main type of metocean event affecting the selected coasts is storm. It is important to note, however, that tropical-like storms in the Mediterranean Sea, also called « Mediterranean hurricanes » or « medicanes », are

occasionally observed. These storms sometimes attain hurricane intensity and can severely affect the coast (Fita et al., 2007). More frequent and higher intensity extreme values are present in the south of France (Patlakas et al., 2020). Classifying the intensity of storms and cyclones as "weak", "moderate" or "severe", Romero and Emanuel (2016) project a higher number of moderate and severe medicanes (at the expense of weak storms) and an increased occurrence of medicanes in the western Mediterranean in the period (2081–2100) compared to the period (1986–2005).


Overseas, the metocean event types are storms and cyclones in the West Indies (Krien et al., 2015; Krien et al., 2017) and Saint-Pierre-et-Miquelon (Catto and Batterson, 2011; Han et al., 2012; Masson, 2014), and storms, cyclones, long swells and tsunamis (moderate exposure) for La Réunion (Lecacheux et al., 2012; Sahal and Morin, 2012; Quentel et al., 2013; Allgeyer et al., 2017) and French Polynesia (Larrue and Chiron, 2010). In Guiana, no phenomenon generating energetic sea states are

identified.

### 3.2.2 Geomorphological configuration of the coast

The geomorphological configuration of each coast is evaluated in Table 6 using three criteria:

- the length of the coast,
- the width of the continental shelf, and
- the convex or concave shape of the coastline.

In the framework of a systemic approach, all of the available knowledge of the study sites is used to assess the surge potential associated with the geomorphological configuration. This knowledge includes information from bibliographical references and data obtained from bathymetric maps provided by the SHOM (Table 6).

In metropolitan France, the geomorphological configurations (i.e. long coastline, wide continental shelf, linear and/or concave shape) generally favor the development of substantial surges. The main exception is the eastern Mediterranean facade, which has convex-shaped coasts without a continental shelf (extension in the sea of the Alpine massif). Another exception is the southern part of the Bay of Biscay, where the concave shape and the length of the coastline favour the surge, while the small width of its continental shelf is a limiting factor.

For the overseas territories, the analysis reveals very contrasting situations:

- Guiana has a broad continental shelf, but its coastline is linear or convex, which is less conducive to surges;
- Saint-Pierre-et-Miquelon is in a situation similar to that of metropolitan France, with a wide continental shelf and the alternation of convex and concave forms of the coast. Although the length of the coastline is smaller, the close proximity of the large island of Newfoundland, which has a 200 km coastline, can contribute to large surges;
- The other islands are far from continents and have continental shelves that are narrow (e.g., Guadeloupe) or non-existent (e.g., La Réunion and French Polynesia), which is, with the short coastline length, a factor limiting the surge magnitude. For La Réunion and French Polynesia, the surges are even more limited by the linear or convex shape of the coastline.

In summary, considering only the geomorphological configuration, the coasts of metropolitan France (except the east of the Mediterranean facade and the Southern part of the Bay of Biscay), French Guiana and Saint-Pierre-et-Miquelon have all the characteristics favoring large surges. Smaller surges are expected on all criteria in La Réunion and French Polynesia, and based on all criteria except the shape of the coast in the West Indies.

**Table 6: Assessment of the surge potential associated with the geomorphological configuration of the study sites (source: Data.shom.fr)**

| | Bibliographical references | Order of magnitude of coastline length (km) | Width of the continental shelf | Shape of the coastline | Surge potential associated with the geomorphological configuration |
|---|---|---|---|---|---|
| English Channel – North Sea | Le Gorgeu and Guitonneau (1954), Hequette (2010), Bardet et al. (2011), Haigh et al. (2011), Weisse et al. (2012), Idier et al. (2012), Maspataud et al. (2013), Hamdi et al. (2014), Vousdoukas et al. (2016), Latapy et al. (2017), Vousdoukas et al. (2017), Hamdi et al. (2018), and DREAL Nord Pas-de-Calais (2024) | 1000 | Very large (>200 km) | Alternation of concave and convex shapes | Very high |
| Bay of Biscay | Allgeyer et al. (2013), Bertin et al. (2014), Hamdi et al. (2014), Bertin et al. (2015), Hamdi et al. (2015), Bulteau et al. (2015), Vousdoukas et al. (2016) and Vousdoukas et al. (2017), Garnier et al. (2018), and Khan et al. (2023). | 1000 | Large (100 to 200 km) | Concave | High |
| Mediterranean | Ullmann et al. (2007), Fita et al. (2007), Campins et al. (2011), Conte and Lionello (2013), Cavicchia et al. (2014), Androulidakis et al. (2015), Vousdoukas et al. (2016), Romero and Emanuel (2016), Vousdoukas et al. (2017), Muis et al. (2020), Elkut et al. (2021), Patlakas et al. (2021), and Toomey et al. (2022). | 1000 | Normal to the West (~ 50 km), nonexistent to the East | Concave to the West and convex to the East | High to the West Low to the East |
| Saint-Pierre-et-Miquelon | Catto and Batterson (2011), Han et al. (2012) and Masson (2014) | 10 | Very large (>200 km) | Alternation of concave and convex shapes | High |
| Guadeloupe | Pagney (1991), Zahibo et al. (2007), Dorville and Zahibo (2010), Lin and Chavas (2012), and Krien et al. (2015) | 5 to 20 | Narrow (<20 km) | Alternation of concave and convex shapes | Low |
| French Guyana | Gratiot et al. (2007), Chevalier et al. (2008), and Thiéblemont et al. (2023) | 1000 | Large (~ 100 km) | Mainly linear or slightly convex | Moderate |
| La Réunion | Lecacheux et al. (2012), Sahal and Morin (2012), Quentel et al. (2013), and Allgeyer et al. (2017) | 20 | Nonexistent | Mainly linear or convex | Very low |
| French Polynesia | Pirazzoli and Montaggioni (1988), Aubanel et al. (1999), Larrue and Chiron (2010), Webb and Kench (2010), Becker et al. (2012), Yates et al. (2013), Le Cozannet et al. (2013), Martinez-Asensio et al. (2019), and Barriot et al. (2023) | 1 to 20 | Nonexistent | Mainly linear or convex | Very low |

### 3.2.3 Other environmental factors influencing the hydraulic conditions

La Réunion and French Polynesia are exposed to long southern swells and will therefore experience direct changes in the swell climate related to the decrease in seasonal sea ice extent.

The islands of the West Indies, La Réunion and French Polynesia are bordered by coral reefs whose degradation by ocean changes resulting from climate change (e.g. increases in water temperature and ocean acidification) may be a major factor in aggravating coastal hazards.

### 3.3 Integrated hazard evolution analysis

The evolution of coastal hazards can be assessed by considering both the quantitative sea level change estimates and the
535 qualitative geomorphological configuration, metocean event and environmental factor analysis for France (Table 7) and its overseas territories (Table 8).

### 3.3.1 Hazard evolution in the European territory of France

**Table 7: Summary of the main factors influencing the hazard evolution on the mainland France's coasts. The data relative to ESLs refer to a present day 100-year ESL. The reference period for the RSL projection is 1995-2014. The reference period for the ESL**
**projection is 1980-2014. The slightly detrimental factors of the evolution of the hazard are in italics, the detrimental factors in bold and the highly detrimental factors in capital and bold. In the first column, these same fonts describe the estimated global evolution of the hazard for each coast.**

| | Surge potential associated with the geomorphological configuration | Reference metocean event | Tidal range | 100-year surge (m) | ΔRSL (m) under SSP5-8.5 | ΔESL (m) under SSP5-8.5 | %ΔESL under SSP5-8.5 | Return period of the present day 100-year ESL under SSP5-8.5 (year) |
|---|---|---|---|---|---|---|---|---|
| **English Channel – North Sea** | **VERY HIGH** | Storm | *Macrotidal (7 to 14 m)* | 1.0 to 1.6 | In 2050 : 0.19<br><br>In 2100 : 0.83 to 0.87 | In 2050 : 0.21 to 0.23<br><br>In 2100 : 0.89 to 1.00 | In 2050 : 3.8 to 5<br><br>In 2100 : 16 to 21.5 | **In 2050 : 20 to 28**<br><br>**In 2100 : 0.73 to 0.87** |
| **Bay of Biscay** | **High to the North**<br><br>Moderate to the South | Storm | *Macrotidal (5 m to 7 m)* | 0.6 to 1.7 | In 2050 : 0.16 to 0.17<br><br>In 2100 : 0.76 to 0.79 | In 2050 : 0,16<br><br>In 2100 : 0.74 to 0.77 | In 2050 : 4.3 to 5.3<br><br>In 2100 : 19.9 to 24.1 | **In 2050 : 33 to 36**<br><br>**In 2100 : 0.63 to 2.6** |
| **WEST MEDITERRANEAN**<br><br>**East Mediterranean** | **High to the West**<br><br>*Low to the East* | Storm and moderate exposure to cyclone | **MICROTIDAL (< 2 m)** | 0.9 to 1.3 | In 2050 : 0.16 to 0.17<br><br>In 2100 : 0.76 to 0.78 | In 2050 : 0,16 to 0.18<br><br>In 2100 : 0.75 to 0.78 | **IN 2050 : 12.5 TO 14.6**<br><br>**IN 2100 : 52.5 TO 63.2** | **IN 2050 : 26 TO 30**<br><br>**IN 2100 : 0.10 TO 0.56** |

By 2100, for the SSP5-8.5 scenario, the increases in RSL and ESL on the three maritime facades are rather similar in absolute terms, with median values between 0.74 and 1.0 m. The effect of these increases is nevertheless significantly different for each maritime facade, depending on the type of metocean event, tidal range, and surge magnitude:

- In the English Channel and the North Sea, the macrotidal range and storm occurrence produce high variability of sea levels. The increase of 0.9 to 1.0 m of the 100-year ESL represents only 16 to 22%. The allowance is therefore relatively low. On the other hand, the increase in the RSL is sufficient for a present day 100-year ESL to have a return period of less than one year (amplification factor greater than 100);

- In the Bay of Biscay, the situation is quite similar, but with a smaller tidal range and surge magnitude (especially on the coast of the Atlantic Pyrenees, where the maximum tides are in the order of 5 m and the 100-year return period surge in the order of 0.6 m). Under these conditions, the 0.75 m increase in the centennial ESL corresponds to a relative increase of 20 to 24%, and above all, a current centennial ESL is likely to occur on average once or twice a year (amplification factor between 50 and 100);

- In the Mediterranean, the situation is clearly aggravated by the very low variability of the sea level (microtidal regime and small surges). As a result, the 100-year ESL increase of 0.77 m corresponds to a large relative increase (greater than 50%) and the present day 100-year ESL is likely to occur between 2 and 10 times per year (amplification factor between 500 and 1000).

By 2050, the 100-year ESL will rise by 5% in the North Channel and the Bay of Biscay. In the Mediterranean, this increase will already be of the order of 20%. However, the evolution of the situation will be especially noticeable in the increased frequency of ESLs (by a factor of 3 to 4).

In addition to this quantitative analysis, the qualitative analysis shows that the geomorphological configurations of the three metropolitan maritime facades (except on the eastern half of the Mediterranean coast and the southern part of the Bay of Biscay) favor the formation of storm surges, by the width of the continental shelves, and the length and shape of the coastlines. On these coasts, a change in the characteristics of storms could therefore lead to a sharp increase in surges. In particular, an increased occurrence and intensity of medicanes in the western Mediterranean, as projected by Romero and Emanuel (2016), would aggravate strongly coastal hazards. Analysis of the surge observed in 2010 during the passage of storm Xynthia in the Bay of Biscay demonstrates the vulnerability of coastal areas to unusual metocean phenomena (Bertin et al., 2012). The storm's low pressure center track propagated to the northeast, contrasting with other storm tracks usually propagating to the east or southeast. Despite its low intensity, this storm caused the large marine submersions in France (Cerema, 2016).

Finally, it should be recalled that the amplification factors related to ESLs (which are expected to reach values between 50 and 1000 in 2100 on the coasts of metropolitan France) must be distinguished from the evolution of extreme events such as

storms, cyclones and tsunamis, whose frequency and intensity is not projected to change as significantly as ESLs. However, even if the frequency and intensity of metocean events remain constant, an increase in the ESL frequency and intensity would increase the effects of these events on the coasts.

### 3.3.2 Hazard evolution in the overseas territories of France

**Table 8: Same as Table 7 for the coastlines of French overseas territories.**

| | Surge potential associated with the geomorphological configuration | Other regional factor of change | Reference metocean event | Tidal range | 100-year surge (m) | ΔRSL (m) under SSP5-8.5 | ΔESL (m) under SSP5-8.5 | %ΔESL under SSP5-8.5 | Return period of the present day 100-year ESL under SSP5-8.5 (year) |
|---|---|---|---|---|---|---|---|---|---|
| **Saint-Pierre (Saint-Pierre-et-Miquelon)** | **High** | *None* | **Storm and moderate exposure to cyclone** | **MICROTIDAL (< 2 M)** | 1 to 2 | In 2050 : 0,19  In 2100 : 0.83 | In 2050 : 0.24  In 2100 : 0.92 | In 2050 : 8.4  In 2100 : 31.8 | **IN 2050 : 18  IN 2100 : 0.3** |
| **POINTE-A-PITRE (GUADELOUPE)** | *Low* | **Coral reef degradation** | **STORM AND CYCLONE** | **MICROTIDAL (< 2 M)** | 1 to 3 | In 2050 : 0,20  In 2100 : 0,90 | In 2050 : 0.22  In 2100 : 0.91 | **In 2050 : 14.1  In 2100 : 58.1** | **IN 2050 : 35  IN 2100 : 0.1** |
| Cayenne (French Guiana) | Moderate | *None* | *Trade winds (without storm)* | **Mesotidal (2 to 4 m)** | **0.4** | In 2050 : 0,20  In 2100 : 0,90 | In 2050 : 0.18  In 2100 : 0.85 | **In 2050 : 10.6  In 2100 : 49.2** | **IN 2050 : 7,5  IN 2100 : 0.1** |
| **POINTE DES GALETS (LA REUNION)** | *Very low* | **Coral reef degradation**  **Decrease in seasonal sea ice extent** | **STORM CYCLONE +LONG SWELL +MODERATE EXPOSURE TO TSUNAMIS** | **MICROTIDAL (< 2 M)** | 1 to 3 | In 2050 : 0,19  In 2100 : 0,92 | In 2050 : 0.16  In 2100 : 0.88 | **In 2050 : 9.3  In 2100 : 52.5** | **IN 2050 : 58  IN 2100 : 0.1** |
| **PAPEETE (FRENCH POLYNESIA)** | *Very low* | **Coral reef degradation**  **Decrease in seasonal sea ice extent** | **STORM CYCLONE +LONG SWELL +MODERATE EXPOSURE TO TSUNAMIS** | **MICROTIDAL (< 2 M)** | 1 to 3 | In 2050 : 0,20  In 2100 : 0,91 | In 2050 : 0.23  In 2100 : 0.95 | **IN 2050 : 17.8  IN 2100 : 74.7** | **IN 2050 : 16  IN 2100 : 0.1** |

In 2100, the quantitative analysis shows that for the SSP5-8.5 scenario:

- The coasts of the West Indies, La Réunion, and French Polynesia are microtidal (<2 m), but the passage of the cyclones can generate large surges (up to 3 m). The allowance (50-75% increase) and amplification (increase in the frequencies of the current centennial ESL by about 1000) of these coasts are similar to those of the Mediterranean coast of France. However, the change in the intensity and frequency of the ESLs must be considered more carefully
in these territories because of the greater damage that tropical cyclones can cause compared to storms (and medicanes) that reach the European territory of France;

- The coast of Saint-Pierre-et-Miquelon is also microtidal (<2 m). While strong winds that regularly blow on these islands can produce large surges, the variability in sea levels remains low because of the microtidal regime. The situation is therefore similar to that of the Mediterranean (including the moderate exposure to cyclones), with a
600 smaller estimated increase in ESLs, as shown by the allowance (about 32%) and amplification (about 300);

- The coast of French Guiana is mesotidal (maximum range of 2.9 m), and the maximum recorded surge is only 0.4 m. The increase in the amplitude and frequency of ESLs should be of the same order of magnitude as that of the Mediterranean coast of France, the West Indies and La Réunion. However, this coast is located close to the equator and is not exposed to metocean events that generate strong swells. The increase in coastal hazards therefore is
605 expected to be less intense than in the other overseas territories.

By 2050, the evolution of the hazard is likely to be perceptible primarily because of the increase in the frequency of ESLs by a factor of 2 for La Réunion, 3 for the West Indies, 5 for Saint-Pierre-et-Miquelon, 6 for French Polynesia, and 13 for French Guiana. The most significant changes are for the territories with low variability in sea levels that are subject to storms and to cyclones (Saint-Pierre-et-Miquelon, La Réunion, the West Indies and French Polynesia, with centennial ESL increases of 8,
9, 14 and 18%, respectively). In this comparison, French Polynesia shows both the greatest increase in frequency and intensity of the 100-year ESL. French Guiana is also expected to experience negative changes in the ESL by 2050, but since it is not exposed to storm waves, this territory is not expected to be exposed to catastrophic metocean events.

In addition, the qualitative analysis shows that the geomorphological configurations of the overseas territories (except for
Saint-Pierre-et-Miquelon and French Guiana) do not favor the formation of storm surges. The situation is aggravated in the West Indies by the degradation of coral reefs and on La Réunion and French Polynesia (islands exposed to cyclones and southern swells) by the degradation of coral reefs and the decrease in seasonal sea ice extent.

Finally, as for the coasts of metropolitan France, it is necessary to distinguish between changes in the frequency and intensity of ESLs and extreme metocean events. Even if extreme metocean events are not expected to increase as significantly as
ESLs, an increase in ESLs will lead to an increase in coastal hazards.

**4 Discussion**

The application of the proposed method leads to increased understanding of coastal hazards, considering the hydrodynamic forcing and the geomorphological context. These results, in conjunction with anticipated coastal responses to the effects of climate change, can shed light on the evolution of the hazard on the terrestrial domain.

**4.1 Influence of local factors affecting the coastline**

Understanding the evolution of coastal hazards requires evaluating slow changes that occur over decadal timescales (e.g. RSL rise, ocean acidification and warming, morphological evolution), in conjunction with extreme events that occur on daily time scales, such as tsunamis, cyclones, storms (Igigabel et al., 2021). However, while some factors can be considered at the regional level (and were therefore included in the proposed method), others can only be considered at finer spatial scales because of the complex interactions between sea level rise and the morphological evolution of coastal areas. In order to assess local hazard evolution, one must study local effects, such as subsidence, especially in deltas (Syvitski, 2008). Second, the coastal zone is a buffer zone where a multitude of processes and feedbacks may also be important:

- in estuaries, changes in coastal morphology (bathymetry, shoreline topography, and anthropogenic development) can influence (positively or negatively) extreme events, including changes in the spread of tidal waves and storm surges (Talke et al., 2020);

- along sandy coasts, coastal morphology is also likely to change as a function of the local morphological and sedimentary characteristics, anthropogenic development, climatic changes (temperature, precipitation and wind), wave conditions, and frequency of events, which has an important impact on the ability of a system to recover between energetic events (Masselink et al., 2016). In addition, wave direction is also an important parameter related to longshore sediment transport, which can change the shape of a coastline (Ruggiero et al., 2010; Casas-Prat and Sierra, 2010);

- in the polar regions, accelerating permafrost thaw is promoting rapid erosion of ice-rich sediments (Lantuit et al., 2011). Melting of ice and associated thaw subsidence may induce instability of various infrastructure components. Arctic RSL rise and sea surface warming have the potential to substantially contribute to this thawing (Lamoureux et al., 2015).

RSL rise can greatly increase coastal hazards, because an increase in water depth, assuming no change in bathymetry, may cause increases in nearshore wave conditions. The coastal impacts of ESLs are largely due to increased wave impacts, potentially driving morphological changes and erosion, as well as coastal protection failure and overwash/inundation (CIRIA et al., 2013, Vousdoukas et al., 2017). Moreover, as indicated by Krien et al. (2017), coastal ecosystems such as mangroves, coral reefs or seagrass beds may not be able to adapt to human activities and climate change (e.g., Waycott et al., 2009; Wong et al., 2014), which could have large impacts on coastal hazards (e.g., Alongi, 2008; Wong et al., 2014; IPCC, 2019).

Ideally, all of these phenomena should be considered in assessing coastal hazard evolution and in defining the adaptation measures for a particular location. While this may be too difficult to achieve for a regional study, it is necessary at the local level.

## 4.2 Understanding hazard evolution on land

On many coasts, accelerated sea level rise is likely to result in permanent submersion of unprotected lowlands. More frequent and intense episodic coastal flooding could also occur with future changes in the wave climate and storm surges. In some locations, this may result in chronic coastline erosion (Ranasinghe, 2016). Thus, depending on the coastline, distinct phenomena may appear, which may be classified, using the proposed method (characterisation of the hazard on a regional scale in the maritime domain) and the insights from existing studies (characterization of phenomena affecting the coastal fringe locally).

By neglecting adaptive actions, hazard evolution may be categorized into one of the following three main scenarios (defined by the sea level variability and the types of metocean events, knowing that other factors also influence the hazard evolution and must be considered subsequently):

- coastlines with low variability in sea levels that are not exposed to metocean events (e.g. French Guiana):  a RSL rise will likely generate a small increase in the risk of rapid submersion by the sea. Nevertheless, rapid flooding by river floods or by intense precipitation remains possible. In addition, the gradual rise of the RSL is likely to cause increased erosion (which, in general, results in coastline retreat) and/or to the progressive and permanent flooding of lowlands (i.e. below sea level);
- coastlines with high sea level variability that are exposed to metocean events (e.g., the Atlantic coastline of metropolitan France):  RSL rise is likely to cause increased erosion and the degradation of protection systems (i.e. anthropogenic structures, such as levees, seawalls, groynes and breakwaters; and natural features such as beaches, dunes, salt marshes, mangroves and coral reefs) over the years, even in the absence of extreme events. During an extreme event (due to the conjunction of a storm, cyclone or tsunami with a high spring tide), the risk may be increased both by the regular weakening of the protection systems and by the increased impacts of waves related to the rise of the RSL;
- coastlines with low sea level variability that are exposed to metocean events (e.g. Saint-Pierre-et-Miquelon, West Indies, La Réunion, French Polynesia): RSL rise is likely to cause, in the short term, the three phenomena described in the previous cases: increased risk of coastal erosion, increased risk of rapid flooding, and permanent flooding of lowlands.

In addition to these three main scenarios, there are coastlines with high variability in sea levels and with relatively low exposure to metocean events, which will likely show slower increases in coastal hazards.

## 5 Conclusion

Changes in sea levels (RSL and ESL) are subject to high uncertainties, at the global scale (economic development, GHG emission scenarios, ice sheet response) and at regional or local scales (response of coastlines to climatic changes and changes in wave climate). Quantified estimates of sea level changes are necessary, but should also be put into perspective with factors describing the state and evolution of the coastal environment. The proposed method for evaluating coastal hazard evolution is based on this principle, emphasizing the importance of considering sea level changes, as well as the geomorphological configuration, metocean event types, and, where appropriate, additional marine environment changes affecting hydrodynamic conditions (in particular, in the tropics or at the poles).

The application of this method to selected French coastlines demonstrated the expected differences in future coastal hazard evolution.

The study of the variability of sea levels with a quantitative approach confirmed the results of previous studies, namely the largest increase of ESLs on coasts with a small tidal range and storm surges. This is the case for the French overseas territories and, in metropolitan France, for the Mediterranean facade. The increase in ESLs, in intensity and frequency, should be limited on the coasts with large tidal ranges (French facades of the Atlantic Ocean, the English Channel and the North Sea).

In addition to the results obtained for ESLs by quantitative approaches, the qualitative approach makes it possible to assess more comprehensively the evolution of coastal hazards.

First, the type of metocean event is essential: coasts not exposed to extreme events (storms, cyclones, long swells or tsunamis), even if they are microtidal (e.g. Guiana), should experience a moderate hazard increase compared to exposed coasts. All other characteristics being equal (geomorphological configurations, environmental changes and tidal conditions), the hazard will increase more rapidly on the coasts exposed to cyclones (the West Indies, La Réunion, French Polynesia, Saint-Pierre-et-Miquelon and the Mediterranean facade) than on the coasts exposed to storms (North Sea, English Channel and Bay of Biscay).

Secondly, certain geomorphological configurations (long maritime facades, concave in shape and bordered by a wide continental shelf) favor the generation of surges during extreme events. These criteria help to identify the coastlines most vulnerable to changes in the characteristics of extreme events (especially their trajectories). This result is important because quantitative approaches are based on climate change scenarios, which therefore represent only part of the possible evolutions. The application of quantitative and qualitative approaches for French territories makes it possible to distinguish

several scenarios: some coastlines benefit from a high variability in sea levels, which is a positive feature to limit the effects of the rise in RSL, but, on the other hand, have geomorphological characteristics favoring surges (e.g. Atlantic, Channel and North Sea coasts). Other shorelines are expected to be disadvantaged by both low sea level variability (small tidal range and moderate surges) and a geomorphological configuration that favors surges (e.g., the western portion of the Mediterranean coastline and the concave shorelines of the West Indies). Finally, some coastlines have the disadvantage of small sea level variability, but their geomorphological configuration does not favor surges (e.g. the eastern part of the Mediterranean coastline). This analysis highlights the need to study not only the water level variability, but also the current and future exposure to metocean events.

Third, the coral reef degradation in the tropics and the decrease in seasonal sea ice extent in the polar regions can also significantly change the nearshore hydrodynamics and impacts on the shoreline.

These results represent a substantial step forward, not only because they are achieved through a comprehensive and systemic approach that requires a step back, but also because the proposed method uses factors for which the uncertainty in the projections is the lowest: the geomorphological configuration of the continental shelf at a regional scale will be invariable over the centuries, regional tidal regime will be stable, climate change impacts on coral reefs (Albright et al., 2018; Hoegh-Guldberg et al., 2018) and seasonal sea ice extent (Forbes, 2011) are fairly well known. However, local climate changes and, more particularly, changes in storm and cyclone trajectories are an area where uncertainties remain particularly high (Seneviratne et al., 2021).

Ultimately, the impacts of sea level changes on human communities are evaluated by considering additionally the local-scale effects on natural features and anthropogenic structures. These analyses aim to predict the types of hazard (shoreline erosion, rapid submersion and/or permanent flooding) that will increase the most in different coastal zones (shoreline erosion, rapid submersion and/or permanent flooding). Shorelines are expected to be marked by a quasi-generalized increase in the risks of erosion of natural features and weakening of anthropogenic structures. However, regional and local-scale differences are expected depending on the submersion type: coastlines with high sea level variability will experience mainly rapid submersion effects, whereas coastlines with low sea level variability will experience mainly permanent flooding of low-lying land, and, rapid submersion when exposed to extreme metocean events.

To go beyond the systemic approach proposed here, a follow-up study could be completed using : (i) a consensus evaluation based on bibliographic studies (IPCC approach) of each of the identified determining factors that have been identified; (ii) or a machine learning approach to explore the added benefits of artificial intelligence (AI). In addition to large-scale hydrodynamic model outputs and other environmental data, analyses may integrate deep learning method outputs. Machine learning can be a powerful tool in reducing greenhouse gas emissions and helping society adapt to a changing climate

(Kaack et al., 2021; Rolnick et al., 2022). AI has already provided interesting results in the field of hydrology (Kumar et al., 2023) and coastal flooding (Tiggeloven et al., 2021; Jones et al., 2023).

In parallel with research on systemic approaches, the assessment of water levels remains an essential topic. Future efforts could benefit from recent developments on ocean modeling like the new generation Global Tide and Surge Model Version 3.0 (GTSMv3.0), which can be used to simulate dynamically tides, storm surges, and changes in MSL, including interaction effects (Muis et al., 2020); or the new global, fully coupled, unstructured model of Mentaschi et al. (2023), coupling waves, storm surge and tides.

## 6 Code availability

The Delft3D-FM code is currently being made available in http://oss.deltares.nl. The WW3 model description is available in: https://polar.ncep.noaa.gov/waves/wavewatch/. The code applied for the non-stationary extreme value statistics (Mentaschi et al., 2016) is available in: https://github.com/menta78/tsEva.

## 7 Data availability

The global ESL data that support the findings of this study are available in the LISCoAsT repository of the JRC data collection (http://data.jrc.ec.europa.eu/collection/LISCOAST) though this link: http://data.jrc.ec.europa.eu/dataset/jrc-liscoast-10012, with the identifiers: https://doi.org/10.2905/jrc-liscoast-10012; PID: http://data.europa.eu/89h/jrc-liscoast-10012

## 8 Author contribution

MI, MY and YD conceived the paper. MV developed the model code and performed the simulations. MI prepared the manuscript with contributions from all co-authors.

## 9 Competing interests

The authors declare that they have no conflict of interest.

## 10 Acknowledgments

The authors would like to thank Xavier Bertin, CNRS research director at the LIttoral ENvironment et Sociétés (LIENSs) laboratory for his comments about the manuscript and for sharing of water height and wave data from the national observation services ReefTEMPS and DYNALIT.

The authors would also like to thank the Naval Hydrographic and Oceanographic Service (in French: Service
hydrographique et océanographique de la Marine or SHOM), as well as Anne-Laure Tiberi-Wadier (Cerema) and Xavier Kergadallan (Cerema) for their contributions to bibliographic research.

This research received no specific grant from any funding agency in the public, commercial, or not-for-profit sectors. All authors declare that they have no conflicts of interest.

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
