# Peer review of "A systemic and comprehensive assessment of coastal hazard changes: method and application to France and its overseas territories"

_Natural Hazards and Earth System Sciences, 2023_

## Referee Comment (RC2)

**Review Report for NHESS-2023-154**

**1  Overall assessment**

The paper titled "A systemic and comprehensive assessment of coastal hazard changes: method and application to France and its overseas territories" is an attempt to, as the title suggests, systemic (e.g, coast as a system) and comprehensive assessment of coastal hazard change (in the context of climate change). The paper goes long discussion about its method, then appears to pull observation from some available repositories (in France), pull climate projection from other studies and proposes to provide a systemic ( or systematic?) and comprehensive assessment of coastal hazard changes. The final result boils down to a table of regions where qualitative/quantitative measurements from the above-mentioned data are put together, and a mixture of subjective and objective opinion is provided.

The objective of this paper is ambitious, and necessary in the context of the risk of multiple coastal hazards, and their unknown/uncertain evolution in the changing climate. I would like to thank the authors to take time to work on this topic. However, unfortunately, after reading the manuscript, I was left with a hollow impression. At the current condition of the manuscript, I do not recommend it's publication in NHESS, and my decision would be to **reject**.

**2  Reasoning for the decision**

Despite my negative decision, below I have tried to provide a relatively broad reasoning of my decision and some ways the work can be improved (in my opinion). I have also marked some smaller matter in the line-by-line comments. I hope it helps the authors to rethink about their approach, analysis, and presentation.

The first one is regarding the physics, and the referencing of the existing knowledge of the physical processes that constitute the hazard. There are countless profound claims regarding storm surge characteristics, tide-surge interactions, contribution of wave setup, link between shoreline and hazard and many more are written, but for which very little or often no reference are provided. For each region in France, for practically each component of hazard in question, there are many available literature that should be cited, but it was not the case. Particularly, the consideration of wave is very highlighted in this paper as a novelty. However, the well outdated (and overused) eqn. of 0.2 Hs [1] is too oversimplification of an important foctor. One way-around would be to get the best assessment from available studies regarding the scaling, if not there are already existing literature that proposes alternative beach-slope dependent formulations (e.g., [2]). The non-linear interaction between various components - tide, surge, wave-setup are now well established (e.g, [3]) and needs to be well thought too. For the overseas areas of France, there are excellent paper exists that uses sophisticated modelling with thousands of cyclones to quantify hazards (e.g., [4, 5]) - they are needed to be consulted. In broader sense, much more effort must be given to harvest the existing knowledge, particularly over France, the case-study of this paper.

Secondly, The organisation of the paper is odd. The method section (section 2) is very long, with a lot of reasoning (which reads like a discussion rather than a method), and always referring to things in France - the study area - which is actually presented afterwards (section 3). It appears like, although I hope my guess is wrong, that the paper was first drafted for France, and then it was re-organise to present as a globally applicable method with an application to France. As such, if the Section 3 and Section 2 are switched, the text makes more sense.

Finally, for a paper this ambitious, no data analysis is done, most of the figures are off-the-shelves, and most of the results are table, which are not often compact, with repeated results. I was looking for a map that summarises these coastal hazards over France, but it was disappointing to not find one. One approach, that might be of interest for the further development of this paper, would be to do a consensus based assessment, where all the contributing factors listed in this paper are assessed based on existing literature. From there to find how consensus the results are - e.g., IPCC approach. Then, how this consensus differs from the results of the current study - which will potentially identify the research gaps in this line of study.

**3 Line-by-Line comments**

1. Abstract: Is appears that "meteocean events" is the main character of your study. Please consider giving a brief definition of what a "metecean event" is in the context of the study.

2. L12: Perhaps you meant "metocean" instead of "meteocean"? In the existing literature, I can only find reference to "metocean" which refers to the combined effect of the meteorologic and oceanographic conditions. If "meteocean" (as currently written in the manuscript) was the term you wanted to introduce, please consider introducing it in this line by incorporating briefly the definition to make its meaning clear (compared to "metocean").

3. L27: Please consider adding a few relevant references to the line "Recent research...".

4. L36: What does "very likely" refers to in this context? Same as IPCC terminology?

5. L40: Why ESL is not sufficient to describe the evolution of the coastal hazards? Please consider brief elaboration of the explanation to come, or provide relevant reference or cases where it was found not enough (e.g., Igigabel et al. 2021 that is cited in L44).

6. L46: Please consider adding a connecting line to indicate for which purpose we need to "First, it is necessary to ...".

7. L122: Does "consequence" here means the same as "impacts" as described in IPCC AR5 WG2 report?

8. L141: "The application of the proposed method ..." for which purpose? It appears that something is missing from this line.

9. L142-144: This statement is interesting and thought-provoking, please elaborate.

10. L184: What would be the 3-maritime facades of France? Perhaps consider adding a bit more somewhere about the coastline of France.

11. L190: Pickering et al. 2012 - The impact of future sea-level rise on the European Shelf tides

12. L201: Is "meteo-oceanic" event is the same as metocean/meteocean event?

13. L213-214: What kind of analysis? How about literature review?

14. L229: "maritime facade" is mentioned again here (directly translated from french façade maritime perhaps? it does not seem to exist in English), please elaborate what it means somewhere in the text.

15. L231: Please consider providing proper journal/article reference (which exists) instead of a generic website from noaa.

16. L234: Please consider adding a real example from published literature.

17. L247: "should" -> "expected to".

18. L250-252: How does these sentences fit to the current discussion of hazard? Please consider rewriting/revising/deleting.

19. L280: Why French coast was chosen to demonstrate the method?

20. Figure 2: Please consider a bit more elaborate caption and add reference to the figure if it is adapted from somewhere else.

21. Figure 2: Please consider putting different colour for so called "marine fecade".

22. L280-300: I do not understand the objective of the paragraph regarding the GMSL projections. Neither in your equation of ESL (eq 3.) nor in the list presented in L270 there is GMSL present.

23. L304: Consider giving a 1-2 line summary of Vousdoukas et al. (2018) framework.

24. L305: Are these projections of waves and storm surges published already? Has there been any bias correction done to CMIP5 data?

25. L308-310: Where are these projection coming from? I do not see a reference here. Is it from CMIP 6 project?

26. Table 2: Please add lon,lat location of the tide gauges. Please also add another column with tidal range.

27. L322: In the "standard classification" are there more than 3 types? Please add a reference to standard classification, like Book of Pugh and Woodworth 2014.

28. Figure 3: The figure contains an incomplete description. The Mediterranean is missing, so is the other french islands. What is type of the data? How it was generated? Model? Altimetry? Tide gauges? Please provide further detail.

29. Table 3: Please consider combining Table 3 with Table 2.

30. L359: Please provide the link for ReefTEMPS and DYNALIT services.

31. L361: Why infragravity waves with Hs ~1m can be superimposed on this set-up? Reference?

32. L363: "This information" -> which information? Which geo-morphological configuration?

33. Table 4-5: Are the results taken directly from Vousdoukas et al. (2018) or reanalyzed? It is not clear to me.

34. L380: Reference for this claim about Mediterranean? Or is it analyzed somewhere in this manuscript? It is not clear.

35. Why Table 5 and Table 7 is separated?

36. Same question as above for Table 4 and Table 6.

37. L405: Repeated, not needed.

38. Table 8: Please add relevant reference to another column. Since it is a "qualitative" assessment, without reference it does not hold enough validity. Adding reference to each cases will also add values to all the past regional studies that are done over these various regions. Same goes for related text, where there appears to be no references currently (L428-446). In addition, it is not clear how these subjective labels are provided - e.g., Very high, High etc.

39. Section 3.2.3: How these factors are taken into account? Any subjective or objective comparison?

40. Why Table 9 and 10 are separated? It seems the tide gauge stations are now aggregated. Why it is so?

41. It is not very clear how in Table 9 and 10, the "important" and "most important" labels are applied. Are they coming from assessment of available literature?

42. L614: I believe GMSL is not taken into account here as global sense, rather it was included into RSL. Is it?

43. L665: How the impact of sea level changes on human communities are evaluated in paper? I do not see it. I do not also see where the "anthropogenic structures" are considered, and how it was considered.

**References**

[1] Jerome Aucan, Ron K. Hoeke, Curt D. Storlazzi, Justin Stopa, Moritz Wandres, and Ryan Lowe. Waves do not contribute to global sea-level rise. *Nature Climate Change*, 9(1):2–2, December 2018.

[2] Sean Vitousek, Patrick L. Barnard, Charles H. Fletcher, Neil Frazer, Li Erikson, and Curt D. Storlazzi. Doubling of coastal flooding frequency within decades due to sea-level rise. *Scientific Reports*, 7(1), may 2017.

[3] Déborah Idier, Xavier Bertin, Philip Thompson, and Mark D. Pickering. Interactions between mean sea level, tide, surge, waves and flooding: Mechanisms and contributions to sea level variations at the coast. *Surveys in Geophysics*, 40(6):1603–1630, jun 2019.

[4] Y. Krien, B. Dudon, J. Roger, and N. Zahibo. Probabilistic hurricane-induced storm surge hazard assessment in guadeloupe, lesser antilles. *Natural Hazards and Earth System Science*, 15(8):1711–1720, aug 2015.

[5] Yann Krien, Bernard Dudon, Jean Roger, Gael Arnaud, and Narcisse Zahibo. Assessing storm surge hazard and impact of sea level rise in the lesser antilles case study of martinique. *Natural Hazards and Earth System Sciences*, 17(9):1559–1571, sep 2017.

---

## Author Comment (AC2)

**Answers to the Review Report for NHESS-2023-154**

**1 Overall assessment**

*The paper titled "A systemic and comprehensive assessment of coastal hazard changes: method and application to France and its overseas territories" is an attempt to, as the title suggests, systemic (e.g, coast as a system) and comprehensive assessment of coastal hazard change (in the context of climate change). The paper goes long discussion about its method, then appears to pull observation from some available repositories (in France), pull climate projection from other studies and proposes to provide a systemic (or systematic?) and comprehensive assessment of coastal hazard changes. The final result boils down to a table of regions where qualitative/quantitative measurements from the above-mentioned data are put together, and a mixture of subjective and objective opinion is provided.*

The main contribution of the paper is the proposed **method**, not the results obtained in the case study. If these results, considered separately or jointly, are also of intrinsic interest, they are mainly presented to demonstrate the interest of our method.

*The objective of this paper is ambitious, and necessary in the context of the risk of multiple coastal hazards, and their unknown/uncertain evolution in the changing climate. I would like to thank the authors to take time to work on this topic. However, unfortunately, after reading the manuscript, I was left with a hollow impression. At the current condition of the manuscript, I do not recommend it's publication in NHESS, and my decision would be to reject.*

We hope that the adaptations we propose below in response to your comments will help to better highlight the contributions and clarify the objectives of our article.

Although we have not tried to make a significant contribution to the quantitative evaluation of the physical phenomena, including extreme water levels and their frequencies, we think that our method of evaluating the evolution of the hazard with a more global approach is still of real interest. The concepts of risk and hazard are at the heart of our reflection, and the apprehension of the hazard for complex systems like the Earth are possible, in our opinion, only if systemic approaches complement the analytical approaches (deterministic or probabilistic). We thus feel like this paper is appropriate for a journal that aims to « embrace a holistic Earth system science approach ».

**2 Reasoning for the decision**

*Despite my negative decision, below I have tried to provide a relatively broad reasoning of my decision and some ways the work can be improved (in my opinion). I have also marked some smaller matter in the line-by-line comments. I hope it helps the authors to rethink about their approach, analysis, and presentation.*

Thank you for clarifying the reasons for your suggestion to reject the paper. Below are our answers.

We also thank you for reviewing the document line by line, as these detailed comments will improve the paper on most of the points you report to us.

*The first one is regarding the physics, and the referencing of the existing knowledge of the physical processes that constitute the hazard. There are countless profound claims regarding storm surge characteristics, tide-surge interactions, contribution of wave setup, link between shoreline and hazard and many more are written, but for which very little or often no reference are provided. For each region*

*in France, for practically each component of hazard in question, there are many available literature that should be cited, but it was not the case. Particularly, the consideration of wave is very highlighted in this paper as a novelty. However, the well outdated (and overused) eqn. of 0.2 Hs [1] is too oversimplification of an important foctor. One way-around would be to get the best assessment from available studies regarding the scaling, if not there are already existing literature that proposes alternative beach-slope dependent formulations (e.g., [2]). The non-linear interaction between various components - tide, surge, wave-setup are now well established (e.g, [3]) and needs to be well thought too.*

The first reference you cite (Aucan et al., 2018) notes the importance of the effect of waves on the sea level. However, our objective is not to evaluate the evolution of a parameter at a precise time, but to improve the overall assessment of coastal hazards in the long term, taking into account that this assessment is carried out using hypothetical scenarios and that there are high uncertainties related to the multiple factors determining the hazard, which are reinforced by the interactions between these factors. In this context, the precise calculation of the effect of waves on the sea level is of relative importance. In particular, concerning the wave set-up estimate, line 172 stated:

« 0.2 $H_s$ is a generic approximation of the wave set-up (U.S. Army Corps of Engineers, 2002). This equation is a conservative estimate of the wave set-up, which mostly depends on the local slope, breaking wave height, and wave period, and may be closer to 10% of the breaking wave height. Larger values may only be observed at steep sandy beaches (e.g. Martins et al., 2022) or steep shore platforms (Sheremet et al., 2014; Lavaud et al., 2022). This expression is used here since the objective of this work is not to improve existing deterministic methods, but to present a systemic and comprehensive method for assessing the evolution of coastal hazards. »

The method for estimating sea levels is given by default and, if desired, the user of the method can produce other estimates. This position may be justified as follows by completing the introduction of Part 2, with three new paragraphs starting with line 142 (and replacing the last sentence of the paragraph) :

**« For the assessment of sea levels, Table 1 shows the projections for the GMSL under the SSP5-8.5 scenario at three time scales: 2050, 2100 and 2300.**

| climate scenario | 2050 Median | 2050 17-83% range | 2100 Median | 2100 17-83% range | 2300 Median | 2300 17-83% range |
|---|---|---|---|---|---|---|
| SSP5-8.5 | + 32 cm | 23 cm to 40 cm | + 84 cm | 61 cm to 110 cm | +385 cm | 230 cm to 540 cm |

**Table 1: GMSL projections for 2050, 2100 and 2300 for the SSP5-8.5 scenario. Median values and ranges for the 17th to 83rd percentiles are shown using the 1986-2005 period as a reference (IPCC, 2019).**

**The GMSL values displayed in Table 1 show high uncertainties in the long term. For example, in 2100, the median value is 0.84 m and the 17th to 83rd percentile range is 0.61 to 1.10 m (the high value is almost double the low value). These GMSL estimates were established for the SSP5-8.5 scenario, but greater differences exist if other scenarios are considered in a complementary manner. Similar (or higher, since other local phenomena must be considered) uncertainties exist for the RSL and the centennial ESL.**

**In conclusion, it is clear that for long-term adaptations, a general description of the evolution of the hazard, including the estimated uncertainties, is more appropriate than a forecast (likely imprecise)**

of the evolution of a parameter at a given date. Accordingly, in this method, we will use conservative assumptions when assessing sea level components. »
In complement:
**At line 157, we will refer to Aucan et al. (2018) and Gomes da Silva et al. (2020) to justify that we do not take wave run-up into account in our method for regional studies.**

**On line 167, equation (3) will be followed by this comment :**

**« Given the strong uncertainties concerning the long-term evolution of the RSL and the geomorphological changes, in the general case it is not strictly necessary, within the framework of this method intended to comprehensively understand coastal hazard changes, to propose a more sophisticated method for estimating ESLs than that proposed by Vousdoukas et al. (2018). Nevertheless, the user of the method should evaluate the advantages of applying another formula if it seems preferable for a particular case study. In particular, Idier et al. (2019) established that in shallow water non-linear interactions between various components - tide, surge, wave set-up can reach several tens of centimeters (on the contrary, non-linear interactions are negligeable in deeper areas). A new generation Global Tide and Surge Model Version 3.0 (GTSMv3.0) was recently developed. GTSMv3.0 can now be used to dynamically simulate tides, storm surges, and changes in MSL, including interaction effects (Muis et al., 2020). »**

Please note that advances in modelling are rapid. While our case studies do not take into account the latest modelling developments, if necessary our method can integrate the results provided by the new models.

**On line 173, after equation (4), we will refer to Stockdon et al. (2006) that propose an alternative beach-slope dependent formulation.**

On line 175, we propose to replace the sentence:
« This expression is used here since the objective of this work is not to improve existing deterministic methods, but to present a systemic and comprehensive method for assessing the evolution of coastal hazards. »
By:
**« I**n the current state of the art, numerical simulations of set-up have been carried out only locally (e.g., Lange et al. (2021), van Ormondt et al. (2021). Such retrospective simulations are computationally expensive (e.g. to simulate accurately set-up, and depending on the local bathymetry and alongshore variations in the wave field, a resolution of ~10-50 m may be needed). In addition : (i) accurate and high resolution wave setup needs good bathymetric data (Stephens et al., 2011) and these are not available in the spatial scales of the study ; (ii) Significant morphological changes are expected in nearshore areas, especially under a rising sea level. Such changes, especially for sandy beaches exposed to waves, can have a significant effect on the wave set-up at different timescales (Ruggiero et al., 2001 ; Thiébot et al., 2012 ; Brivois et al., 2012). Thus, at large spatial (e.g. global) and long temporal (e.g. 21st century) scales, simplified models that like of Voudoukas et al. (2018) may provide a first estimate of the expected wave set-up. »**

We feel as though the work of Vousdoukas et al. (2018) framework is an appropriate choice for the desired spatial and temporal scales in this study.

*For the overseas areas of France, there are excellent paper exists that uses sophisticated modelling with thousands of cyclones to quantify hazards (e.g., [4, 5]) - they are needed to be consulted. In broader sense, much more effort must be given to harvest the existing knowledge, particularly over France, the case-study of this paper.*

In principle, since we defend a systemic approach in addition to the analytical approach, our method should promote the use of all the knowledge available in a territory. We will therefore include additional relevant references in the case studies (these references are presented in our responses to your detailed comments). You will understand however that our first intention for the presentation of the nine case studies, is to show the form that can take the results (our second intention is to show the differences that may appear in the results obtained on the different sites).

*Secondly, The organisation of the paper is odd. The method section (section 2) is very long, with a lot of reasoning (which reads like a discussion rather than a method), and always referring to things in France - the study area - which is actually presented afterwards (section 3). It appears like, although I hope my guess is wrong, that the paper was first drafted for France, and then it was re-organise to present as a globally applicable method with an application to France. As such, if the Section 3 and Section 2 are switched, the text makes more sense.*

Since the primary objective of the paper is to present a method, section 2 is devoted a detailed presentation of the approach, explaining the underlying concepts (that of risk in particular) and the assessment principles (qualitative systemic approach to accompany the quantitative analytical approach, how to take into account the high uncertainties generated by climate change on the various components of the hazard).
The order and format of presentation of the data associated with the different sites is an integral part of the application of our method. Therefore, it is necessary to maintain the structure of the article "introduction - method - data of study sites and results - discussion - conclusion", even if a plan "introduction - data of study sites - method - results - discussion - conclusion" would have been possible without this constraint.

In response to your comment (and the comments of the first reviewer), we will present the method more concisely. These changes will reduce the number of references to metropolitan France (only one reference will remain (line 215)) and rebalance with references to other regions of the world (e.g. West Indies (line 222), United States (line 223), La Reunion (line 228), Polar regions (line 253)).

*Finally, for a paper this ambitious, no data analysis is done, most of the figures are off-theshelves, and most of the results are table, which are not often compact, with repeated results. I was looking for a map that summarises these coastal hazards over France, but it was disappointing to not find one.*

To give all the necessary follow-up to this comment (and the previous one), we propose to replace the last two paragraphs of the introduction (line 90-99) with the following paragraphs which specify the objectives of the study and its protocol of elaboration:

**« The objective of this paper is to present a comprehensive method to assess the evolution of coastal hazards at regional scales in the context of climate change. The proposed systemic method emphasizes the need to focus on the analysis and interpretation of the modelling results, by putting them into perspective with respect to the biophysical conditions (both current and forecasted).**

**The development of the method is the result of an empirical process, considering diverse situations. This process has the advantage of demonstrating the wide applicability of the proposed method. The case studies used to develop the method are all in France and French territories, but are located in different latitudes (equator, tropics and temperate zones), exposed to different climates, and characterized by different geomorphological configurations (including continental or island). It therefore becomes necessary to consider the qualitative « additional factors » proposed in the method that make it possible to assess the evolution of hazards on most coastlines in the world.**

For the same purpose of ensuring that the method can be applied for operational purposes, the use of freely accessible data has been promoted. However, applications of this method should include a thorough bibliographic analyse or even additional investigations at the chosen case study sites. It is important to emphasize here that the application of the method depends strongly on the available data, and therefore it is necessary to gather the best data for each site. Since the quantity and quality of data is not the same everywhere, the uncertainties in the results will also vary, and must be addressed.

Following the application of the method to the nine case studies, the obtained results can contribute to improving predictions of the evolution of different types of coastal hazards (shoreline erosion, rapid submersion and/or permanent flooding). »

In addition, we can merge and refine some tables to present the data in a more compact format (see our responses to your detailed comments).

Regarding the representation of the evolution of hazards in France and French territories in the form of a map, this would not bring additional information compared to the text, in particular since regional hazard analyses should extend to the local scale, as indicated in Section 4.

*One approach, that might be of interest for the further development of this paper, would be to do a consensus based assessment, where all the contributing factors listed in this paper are assessed based on existing literature. From there to find how consensus the results are - e.g., IPCC approach. Then, how this consensus differs from the results of the current study - which will potentially identify the research gaps in this line of study.*

Extending the research by the consensus method could be interesting, provided that consensus can be reached, which is not obvious given the diversity of coastal contexts. In all cases, we cannot present in the same paper both our method developed on an empirical basis and other methodological principles developed by consensus. On this point, **we propose to include in the conclusion as a research perspective your suggestion of a consensus evaluation based on bibliographic studies on each of the determining factors.**

Note that two questions you ask in the detailed comments are fundamental for our paper:

-in line 40 : *Why ESL is not sufficient to describe the evolution of the coastal hazards?*

-in line 250-252 : *How does these sentences [relative to the functional collapse of coral reefs] fit to the current discussion of hazard ?*

We plan to respond to this in the introduction of the paper, starting at line 45 (where you correctly pointed out that a transition was missing):

**« To answer this problem [hazard cannot be represented by a single parameter: the maximum water level reached during an event], it should first be recalled that during a metocean event, the phenomena of flooding and coastal erosion are not only determined by the maximum sea level, but also by coastal waves and currents, and overtopping and overflow discharges over flood protection structures (Formentin et al., 2018; Igigabel et al., 2022). Estimating these discharges and hydrodynamic conditions for the duration of an event requires a good understanding of the physical phenomena that generate the hazard. Retrospective analyses of events help to understand correctly the mechanisms that cause the observed flooding or erosion. For example, by simulating Total Water**

Levels (TWLs) along the Bight extending from North Carolina to Florida during three historical Tropical Cyclones (TCs) with similar tracks, Hsu et al. (2023) found that the magnitude and duration of the increase in TWLs and wind waves are influenced by TC intensity, translational speed and distance from shore. In particular, these authors established that a decrease in TC translation speed led to longer exceedance durations of TWLs, which may result in higher impacts.

Unfortunately, it is not possible to predict the physical characteristics of future events, nor to assess corresponding hydrodynamic conditions. To compensate for this, probabilistic approaches have been developed. For example, using a large number of synthetic hurricanes that consider the natural variability in hurricane frequency, size, intensity and track, Krien et al. (2015) infer for the archipelago of Guadeloupe 100-year and 1000-year surge levels. Following the same principle, Krien et al. (2017) estimate 100-year surge levels in Martinique for the present climate or considering a potential sea level rise. These results help to determine the necessary levels of protection structures in the short and medium term. However, a single parameter (the maximum water level) is not enough to characterize the hazard and define all the crisis management measures, particularly when water levels exceed the level of protection or when protection structures fail (for example, breaches in levees or dunes). In addition, the accuracy of these estimates decreases in the long term due to: (i) high uncertainties in sea levels beyond 2050 (IPCC, 2019); (ii) the increase in the proportion of high-intensity cyclones worldwide (Masson-Delmotte et al., 2021); and (iii) environmental instability, notably because of geomorphological (e.g., subsidence, coastline retreat) and biological (e.g., degradation of coral reefs and mangroves) changes. These changes modify hydrodynamic and hydrosedimentary processes on the coast both in the long term and during individual events.

With the aim of making progress in the global assessment of coastal hazards in the long term, the guiding principle of this paper is to promote the use of the latest advances in research on changes in metocean events and water levels, while also encouraging the study of other factors whose evolution are more predictable than storm surge and that may be equally important, namely: tidal regimes, geomorphological settings and environmental changes (particularly those modifying hydrodynamic conditions at the coast).

Although water levels are not the only parameter to consider, it is necessary to begin by clarifying the definitions of the different levels to which we will refer regularly. »

**3 Line-by-Line comments**

*1. Abstract: Is appears that "meteocean events" is the main character of your study. Please consider giving a brief definition of what a "metecean event" is in the context of the study.*

For clarity, by convention, we will use " metocean events" throughout the document to generically name storms, cyclones, and tsunamis (even if the former have no meteorological components).

*2. L12: Perhaps you meant "metocean" instead of "meteocean"? In the existing literature, I can only find reference to "metocean" which refers to the combined effect of the meteorologic and oceanographic conditions. If "meteocean" (as currently written in the manuscript) was the term you wanted to introduce, please consider introducing it in this line by incorporating briefly the definition to make its meaning clear (compared to "metocean").*

Idem.

*3. L27: Please consider adding a few relevant references to the line "Recent research. . . ".*

We will cite Cazenave and Llovel (2010), Cazenave and Le Cozannet (2013), Hamlington et al. (2020) and Fox-Kemper et al. (2021).

*4. L36: What does "very likely" refers to in this context? Same as IPCC terminology?*

Yes, the probability is between 90 and 100%.

*5. L40: Why ESL is not sufficient to describe the evolution of the coastal hazards? Please consider brief elaboration of the explanation to come, or provide relevant reference or cases where it was found not enough (e.g., Igigabel et al. 2021 that is cited in L44).*

Response provided at the end of the general comments.

*6. L46: Please consider adding a connecting line to indicate for which purpose we need to "First, it is necessary to . . . ".*

Response provided in the general comments.

*7. L122: Does "consequence" here means the same as "impacts" as described in IPCC AR5 WG2 report?*

Yes.

*8. L141: "The application of the proposed method . . . " for which purpose? It appears that something is missing from this line.*

OK, the text will be edited as follows:

**« To determine the most relevant scenario and identify the most appropriate time horizon, an exposure and vulnerability analysis and an assessment of the duration of the project are required. These reflections are an essential prerequisite for the application of the proposed method. »**

*9. L142-144: This statement is interesting and thought-provoking, please elaborate.*

Response provided at the end of the general comments.

*10. L184: What would be the 3-maritime facades of France? Perhaps consider adding a bit more somewhere about the coastline of France.*

OK, we can improve the localisation of these facades.

*11. L190: Pickering et al. 2012 - The impact of future sea-level rise on the European Shelf Tides*

OK, thank you for this reference.

*12. L201: Is "meteo-oceanic" event is the same as metocean/meteocean event?*

Yes, and we will prefer « metocean » throughout the paper.

*13. L213-214: What kind of analysis? How about literature review?*

We will replace the sentence

« First, the analysis of storm surges computed from the national REFMAR database reveals that they are controlled not only by storm tracks but also by the width of the continental shelf and the presence of shallow waters. »

By :

« **Storm surges are controlled not only by storm caracteristics, such as for tropical cyclones (TC) the TC intensity, the distance to the TC eye, the TC heading direction and the TC translation (Hsu et al., 2023), but also by the width of the continental shelf and the presence of shallow waters (kennedy et al., 2012).** »

**In the next paragraph we will refer to Krien et al. (2015) and Krien et al. (2017) to support the fact that « on the islands of the West Indies, the storm surges rarely exceed 3 m ».**

*14. L229: "maritime facade" is mentioned again here (directly translated from french façade maritime perhaps? it does not seem to exist in English), please elaborate what it means somewhere in the text.*

« Facade » is a word also used by English native speakers. **The localisation of these facades on a map will definitely help to understand the text.** Thank you.

*15. L231: Please consider providing proper journal/article reference (which exists) instead of a generic website from noaa.*

**We will refer to Flather (2001), Rego and Li (2010) and Kennedy et al. (2012).**

*16. L234: Please consider adding a real example from published literature.*

Idem.

*17. L247: "should" -> "expected to".*

To avoid redundancy with « expected », we propose « may ».

*18. L250-252: How does these sentences fit to the current discussion of hazard? Please consider rewriting/revising/deleting.*

As indicated in the answers to general comments, biological changes should be taken into account in the assessement of hazard changes.

In complement, we could add in the discussion (L. 581), as indicated in Krien et al. (2017) : « **Moreover, coastal ecosystems such as mangroves, coral reefs or seagrass beds may not be able to adapt to climate change (e.g., Waycott et al., 2009; Wong et al., 2014), which could have large impacts on coastal hazards (e.g., Alongi, 2008; Wong et al., 2014).** »

*19. L280: Why French coast was chosen to demonstrate the method?*

Response provided in the general comments.

*20. Figure 2: Please consider a bit more elaborate caption and add reference to the figure if it is adapted from somewhere else.*

OK.

*21. Figure 2: Please consider putting different colour for so called "marine facade".*

OK, good idea. Thank you.

*22. L280-300: I do not understand the objective of the paragraph regarding the GMSL projections. Neither in your equation of ESL (eq 3.) nor in the list presented in L270 there is GMSL present.*

OK, you're right. In an earlier version of the paper, this text was positioned at line 144.

Here, we propose « **The choice of the scenario and the time scales should take into account the existence of the high value assets at the considered coasts (coastal cities, port and industrial facilities) and the strong uncertainties about the contributions of ice caps to the rise in water levels (Bamber et al., 2019; Dayan et al., 2021). For these two reasons, we use the SSP5-8.5 scenario.** »

And the rest of the paragraph will be moved to section 2 to explain the high uncertainties in sea level changes.

*23. L304: Consider giving a 1-2 line summary of Vousdoukas et al. (2018) framework.*

We propose to replace:

« Projections of waves and storm surges were based on hydrodynamic simulations driven by atmospheric forcing from six Coupled Model Intercomparison Project Phase 5 (CMIP5) climate models. »

By:

« **In this framework, baseline values are based on global reanalyses of waves and storm surges. Then CMIP5 models are used to estimate future relative changes to the meteorological tide, and these are applied to estimate the future values. Lastly, meteorological water level changes, astronomical tide and sea level rise are combined to produce the ESL values.** »

*24. L305: Are these projections of waves and storm surges published already? Has there been any bias correction done to CMIP5 data?*

No bias correction is needed since we are using the CMIP5 simulation results only to obtain relative changes. In addition the results have been validated in the following articles:

Vousdoukas, M.I., Mentaschi, L., Voukouvalas, E., Verlaan, M., Jevrejeva, S., Jackson L. P. & Feyen, L., 2018. Global probabilistic projections of extreme sea levels show intensification of coastal flood hazard. Nat Commun 9, 2360. https://doi.org/10.1038/s41467-018-04692-w.

Mentaschi, L., M. I. Vousdoukas, E. Voukouvalas, A. Dosio, and L. Feyen, 2017. Global changes of extreme coastal wave energy fluxes triggered by intensified teleconnection patterns, Geophys. Res. Lett., 44, 2416–2426, doi:10.1002/2016GL072488.

Vousdoukas, M. I., Mentaschi, L., Voukouvalas, E., Verlaan, M., and Feyen, L., 2017. Extreme sea levels on the rise along Europe's coasts. Earth's Future, 5, 304–323. doi:10.1002/2016EF000505.

*25. L308-310: Where are these projection coming from? I do not see a reference here. Is it from CMIP 6 project?*

As indicated, it is CMIP5 and not CMIP6.

*26. Table 2: Please add lon,lat location of the tide gauges. Please also add another column with tidal range.*

**Good idea to add latitude and longitude.**

Tidal ranges are presented in 3.1.2. To keep the structure of the section and avoid redundancy, we prefer not to add this information in Table 2.

| | Latitude | Longitude | Projection of RSL rise (m) | |
|---|---|---|---|---|
| | | | 2050 | 2100 |
| Calais | 50.972122801049395 | 1.8400588679271384 | 0.19 | 0.86 |
| Le Havre | 49.485774012966544 | 0.0897840202510471 | 0.19 | 0.87 |
| Saint-Malo | 48.641170797005124 | -2.0313888026241402 | 0.19 | 0.87 |
| Brest | 48.368792381060274 | -4.4887286031866935 | 0.19 | 0.83 |
| La Rochelle | 46.14879125637811 | -1.1691588857635562 | 0.16 | 0.76 |
| Saint-Jean-de-Luz | 43.39842588942395 | -1.676715829943722 | 0.17 | 0.79 |
| Port Vendres | 42.52411089711506, | 3.1143835886292806 | 0.16 | 0.76 |
| Sète | 43.39330679508365 | 3.699492031443802 | 0.16 | 0.76 |
| Marseille | 43.29569227471328 | 5.352448215467127 | 0.17 | 0.78 |
| Saint-Pierre (Saint-Pierre-et-Miquelon) | 46.786272435631275 | -56.16190646722868 | 0.19 | 0.83 |
| Pointe-à-Pitre (Guadeloupe) | 16.23300045952869 | -61.53571250198115 | 0.20 | 0.90 |
| Cayenne (French Guiana) | 4.93572687841612 | -52.340676954198194 | 0.20 | 0.90 |
| Pointe des Galets (La Réunion) | -20.936178918145654 | 55.280686667086655 | 0.19 | 0.92 |
| Papeete (French Polynesia) | -17.53479287238126 | -149.58674796473545 | 0.20 | 0.91 |

*27. L322: In the "standard classification" are there more than 3 types? Please add a reference to standard classification, like Book of Pugh and Woodworth 2014.*

**We will refer to Masselink and Short (1993).**

*28. Figure 3: The figure contains an incomplete description. The Mediterranean is missing, so is the other french islands. What is type of the data? How it was generated? Model? Altimetry? Tide gauges? Please provide further detail.*

The objective of our article is not to detail as much as possible the quantitative assessment of the different components of the water level. As for the wave set-up, the goal is to provide the order of magnitude.

The map is provided by the SHOM, which produces the reference maritime and coastal geographic information in France. It aims at illustrating tidal ranges along macrotidal coastlines. That's why the information is limited to English Channel and the Atlantic. The other coasts are microtidal or mesotidal.

*29. Table 3: Please consider combining Table 3 with Table 2.*

To keep the structure of the section, we would prefer not to combine Table 3 with Table 2.

*30. L359: Please provide the link for ReefTEMPS and DYNALIT services.*

DYNALIT: https://www.dynalit.fr/ and REEFTEMPS: https://www.reeftemps.science/

*31. L361: Why infragravity waves with Hs ~1m can be superimposed on this set-up? Reference?*

Bertin et al. (2020) showed that infragravity waves of Hs~1.5 m were superimposed to a surge (wind + wave setup) of 0.5 to 1.0 m.

Baumann et al. (2017) showed that IG waves could reach 2 m during storm Hercules.

In Truc Vert, Ruessink (2010) measured IG waves of Hs~1.5 m during storm Johanna, he did not provide estimates of wave setup but Nicolae-Lerma et al. (2017) provided estimates of wave setup up to 0.8 m for this storm.

To keep the text short, we propose to add only the reference to Bertin et al. (2020).

*32. L363: "This information" -> which information? Which geo-morphological configuration?*

« This information confirms » can be replaced by « These observations confirm ».

*33. Table 4-5: Are the results taken directly from Vousdoukas et al. (2018) or reanalyzed? It is not clear to me.*

As indicated in lines 304, 366 and 386, the results are taken directly from the framework developped by Vousdoukas et al. (2018). The reasons for this choice are explained in the general comments and will be transcribed in the paper.

*34. L380: Reference for this claim about Mediterranean? Or is it analyzed somewhere in this manuscript? It is not clear.*

**We will refer to Vitousek et al. (2017), since this reference is cited concerning this subject in the introduction.**

*35. Why Table 5 and Table 7 is separated?*

The idea is to analyse separately mainland France and overseas territories. If you want, we can merge the tables and adapt the texts accordingly.

*36. Same question as above for Table 4 and Table 6.*

Idem.

*37. L405: Repeated, not needed.*

**OK.**

*38. Table 8: Please add relevant reference to another column. Since it is a "qualitative" assessment, without reference it does not hold enough validity. Adding reference to each cases will also add values to all the past regional studies that are done over these various regions. Same goes for related text, where there appears to be no references currently (L428-446). In addition, it is not clear how these subjective labels are provided - e.g., Very high, High etc.*

**The diversity of situations does not really make it possible to set up a calibration and weighting system. Here our method aims to help experts to make a structured judgment on the « surge potential associated with the geomorphological configuration » based on a list of criteria. We agree that experts should also be encouraged to integrate past study results in their jugments. To this end past regional studies should indeed appear in Table 8. Here are the references we propose to cite on the different coastlines considered for our case studies.**

**For the three metropolitan French facades :**

[revised manuscript text omitted]

*39. Section 3.2.3: How these factors are taken into account? Any subjective or objective comparison?*

In the proposed method, we suggest checking that such factors (changes in the swell climate related to the decrease in seasonal sea ice extent, and coral reefs degradation) are not likely to significantly change the hydraulic conditions in the long term. It is difficult to make comparisons with the influence of other factors (e.g. geomorphological configuration, reference metocean event, tidal range, frequency and intensity of extreme climate events).

*40. Why Table 9 and 10 are separated? It seems the tide gauge stations are now aggregated. Why it is so?*

The goal is to avoid too much information in the same table.

The aggregated data provide regional conclusions. If conclusions are sought at a more local scale, as explained in Section 4, it is necessary to extend the investigations.

*41. It is not very clear how in Table 9 and 10, the "important" and "most important" labels are applied. Are they coming from assessment of available literature?*

Indeed, it would be better to qualify the factors by the terms "slightly detrimental ", " detrimental " and "highly detrimental".

**As for Table 8, in Table 9 and 10 the diversity of the situations does not really make it possible to set up a calibration and weighting system.**

**In our method, each factor should not be considered independently. On the contrary, the joint effects of the different factors should be assessed to understand the dynamics of the system.**

**This is the main reason why our method can be called « systemic », in reference to the study of a system (as you indicated, in our case, the coast). However, our method should not be called « systematic ». Whereas it provides a framework for analysis that incorporates multiple factors, the expert in charge of the study of a particular site should nevertheless form his own opinion by considering these factors together. In summary, our method aims to help experts to make a structured judgment.**

**We propose to insert this explanation at the end of the presentation of the method (line 279).**

42. L614: I believe GMSL is not taken into account here as global sense, rather it was included into RSL. Is it?

**You're right.**

*43. L665: How the impact of sea level changes on human communities are evaluated in paper? I do not see it. I do not also see where the "anthropogenic structures" are considered, and how it was considered.*

**Anthropogenic structures are mentioned explicitly in section 4.1 for estuaries and polar regions. We can also mention them for sandy coasts.**

**Additional references**

Alongi, D. M.: Mangrove forests: resilience, protection from tsunamis, and response to global climate change, Estuar. Coast. Shelf S., 76, 1–13, 2008.

Aucan J., Hoeke R. K., Storlazzi C. D., Stopa J., Wandres M., and Lowe R.: Waves do not contribute to global sea-level rise. Nature Climate Change, 9(1):2–2, December 2018.

Bertin X., Martins K., de Bakker A., Chataigner T., Guérin T., et al.: Energy Transfers and Reflection of Infragravity Waves at a Dissipative Beach Under Storm Waves. Journal of Geophysical Research. Oceans, 125 (5), pp.e2019JC015714, 2020.

Brivois O., Idier D., Thiébot J., Castelle B., Le Cozannet G., Calvete D.: On the use of linear stability model to characterize the morphological behaviour of a double bar system. Application to Truc Vert beach (France). CR Geosci 344:277–287. https ://doi.org/10.1016/j.crte.2012.02.004, 2012.

Cazenave, A. and Llovel, W.: Contemporary sea level rise. Ann. Rev. Marine Sci. 2, 145‑173, 2010.

Cazenave, A. and Le Cozannet, G.: Sea level rise and its coastal impacts, Earth's Future, 2, 15–34. doi:10.1002/2013EF000188, 2013.

Flather, R.A.: Storm surges. In: Steele, J., Thorpe, S., Turekian, K. (Eds.), En- cyclopedia of Ocean Sciences. Academic, San Diego, California, pp.2882–2892, 2001.

Formentin, S.M. and Zanuttigh, B.: A new method to estimate the overtopping and overflow discharge at over-washed and breached dikes. Coast. Eng., 140, 240–256. https://doi.org/10.1016/j.coastaleng.2018.08.002, 2018.

Fox-Kemper, B. et al., 2021. Ocean, cryosphere and sea level change. In IPCC AR6 (eds Fox-Kemper, B. et al.) 1211–1362. Cambridge University Press.

[revised manuscript text omitted]

**References cited in this document that will not be included in the paper**

Baumann J., Chaumillon E., Bertin X., Schneider J.-L., Guillot B., Schmutz M. : Importance of infragravity waves for the generation of washover deposits, Marine Geology, Volume 391, Pages 20-35, ISSN 0025-3227, https://doi.org/10.1016/j.margeo.2017.07.013, 2017.

Nicolae-Lerma A., Pedreros R., Robinet A., Sénéchal N.: Simulating wave setup and runup during storm conditions on a complex barred beach, Coastal Engineering, Volume 123, Pages 29-41, ISSN 0378-3839, https://doi.org/10.1016/j.coastaleng.2017.01.011, 2017.

Ruessink, B. G.: Observations of turbulence within a natural surf zone. Journal of Physical Oceanography 40, 2696–2712, 2010.

---

## Author Response (AR1)

**Dear reviewers and community members,**

We sincerely thank you for taking part in the review process of our article. The assessment of coastal hazard changes is a complex subject that requires crossing points of view. This was the rationale for bringing together specialists from different disciplines for the initial writing. Thanks to your insights on the fields of coastal risk management, but also hydrology and water resource management, we were able to specify our systemic method and the conditions of its implementation.

The responses to your reports are gathered below (these responses were updated to be consistent with the revised version). Your recommendations and suggestions appear in italics and are underlined. Our answers are in bold when they induce changes to our article and in normal characters when they relates to supplementary comments.

For a better traceability of the changes, in accordance with the editor's instructions, we send you in addition to the revised version of our article, a marked-up version of our revised manuscript. As you can see, the changes are very substantial and we hope that the revised version will give you full satisfaction.

**Anonymous Referee #1**

*1. How the author correlates the hazard with climate change?*

The impact of climate change on coastal hazards is the result of the interaction between climate and ocean variables (which may be highly correlated with one another). For this reason, we considered that « given the multiple effects of climate change on sea levels (GMSL and RSL), atmospheric conditions, and wave conditions, the evolution of the factors and parameters can only be assessed using global development hypotheses » (line 136).

*2. How the frequency variations of extreme sea levels (ESL) was studied by the author?*

General principles are presented in the introduction (line 73-89) and a more detailed presentation is given in the method (line 159 – 176). The method focuses on changes in the magnitude and frequency of occurrence of the present 100-year ESL ($ESL_{100}$), following Vousdoukas et al. (2017).

*3. Why author have selected France for study purpose?*

They are two main reasons.

The first reason is explained at the end of the introduction (line 95-99): "France and its overseas territories are located in different latitudes (equator, tropics and temperate zones), exposed to different climates, and characterized by different geomorphological configurations (continental or island)." Indeed, the application of this method to selected territories allowed to highlight the expected differences in the future evolution of coastal hazards.

The second reason is that the authors had a good access to French territory data:

- bathymetric maps (provided by the SHOM)
- analysis of storm surges computed from the national REFMAR database (provided by the SHOM)
- Continuous water height and wave measurements from the French national observation services ReefTEMPS and DYNALIT (provided by the LIttoral ENvironment et Sociétés (LIENSs) laboratory)
- Scientific reports relative to extreme sea levels and morphodynamics along French coasts (provided by Cerema).

*4. Have authors compared their work with the other researchers?*

First, we have considered the work of other researchers on physical and biological changes due to climate change on the ocean and the coast (sea level, wave, geomorphology, coral reefs, sea ice extent, etc.). The main references cited in this field are :

- the Concepts and Terminology for Sea Level (e.g. Gregory et al., 2019)
- the mechanisms generating sea-level rise (e.g. Cazenave and Le Cozannet, 2013 ; Frederikse et al., 2020 ; Haigh et al., 2020; Talke et al., 2020) and storm surges (e.g. Bertin et al., 2012; Dodet et al., 2019; Calafat et al., 2022)
- sea level rise projections (e.g. Lowe et al., 2010 ; Church et al., 2013; Slangen et al., 2017; Bamber et al., 2019; Dayan et al., 2021 )
- Height and frequency variations of ESL (Hunter, 2012; Buchanan et al., 2016; Vitousek et al., 2017; Vousdoukas et al., 2017; Vousdoukas et al., 2018)

- Changes in the wave directional frequencies (Casas-Prat and Sierra, 2010) and shoreline changes (Ruggiero et al., 2010; Forbes, 2010 ; Lantuit et a., 2011 ; Lamoureux et al., 2015, Masselink et al., 2016 ; Ranasinghe, 2016; Lavaud et al., 2022 ; Martins et al., 2022)
- Coral Reefs changes (Hoegh-Guldberg, 2007 ; Albright et al., 2018)
- How to translate hazards into Impacts or Risks.

Second, we have studied how these biophysical changes were integrated in coastal hazard and risk analysis. It appears that some authors adopted an analytical approach that is convenient for the issue they are addressing, for example the implications of extreme coastal water levels for potential coastal overtopping (Almar et al., 2021). But in many cases, as indicated in the introduction (line 40-44), the analytical approach should be completed with a more systemic approach. For example, to assess future flood damage in the major coastal cities (see e.g. Nicholls et al., 2008; Hallegatte et al., 2013; Rassmussen et al., 2022) or on the coast worldwide (Tiggeloven et al., 2020), the hazard should not be represented only by ESLs. In particular, when the objective is to assess risk in a comprehensive way, further reflection is needed on the definition of hazard in the context of climate change.

Therefore, we considered that a shortcoming existed in the scientific litterature and we proposed a systemic assessment method, as indicated in the abstract, "to analyze coastal hazard changes at regional scales, integrating parameters influencing sea-levels, as well as factors describing the geomorphological context (length and shape of the coast, width of the continental shelf), meteocean events (storms, cyclones and tsunamis), and the marine environment (e.g., coral reef state and sea ice extent)".

*5. Through some light regarding need of the study?*

**In addition to justifying our proposal scientifically (as indicated above, filling a gap in the state of the art), we could actually add in the text an explanation of the need to disseminate our method in response to operational needs. We propose to add the following sentence, at the end of the introduction (after line 95):**

***The proposed systemic method emphasizes the need to focus on the analysis and interpretation of the modelling results, by putting them into perspective with respect to the biophysical conditions (both current and forecasted).***

*6. Steps are explained in a detailed manner. It is requested to minimize the same.*

**For step 1, we propose to shorten three paragraphs (line 178-192). They would be replaced by :**

***In general, tidal simulations show no significant impacts of RSL rise on tidal amplitude during the 21st century at regional scales, although this does not exclude potential local effects (Haigh et al., 2020 ; Idier et al., 2017). Given the strong uncertainties on RSL rise, we will assume here that the tides are in a steady state and, consequently, that the tidal range does not change the allowance.***

**For step 2, we propose to shorten one paragraph relative to meteocean event types (line 204-209). It would be replaced by :**

***The increase in wave damage could also be assessed, considering local changes in significant wave height (average height of the highest one-third of the waves in a given sea state). However, trends in coastal wave climates are reported, with a low confidence level, in the IPCC (2019) report. Therefore, these trends will not be explored in detail: the focus will be on the strong differences that already exist between the wave climates of the maritime facades.***

**For step 3, we propose to delete one paragraph (261-267).**

It will avoid redundancy and information that are not essential for the implementation of the method.

Paragraphs justifying the choice of qualitative parameters would be kept unchanged.

*7. What is the importance of geomorphology in your study?*

The main interest of our study is to consider geomorphology in conjunction with the other factors determining coastal hazards, which appears in our conclusions (lines 640 to 663).

*8. Give citations wherever required.*

**It is interesting, as you suggest, to refer to work on water resource management and flood risk in the continental domain. This enriches the reflection on the methods, even if it goes a bit beyond the scope of the current work. Additional quotations have been added (those you mentioned and two others relating to regional sea-level change and the use of artificial intelligence in the field of coastal risks). On the other points, we made sure to systematically cite the authors we referred to.**

*9. Add below mentioned papers and cite them in the text:*

**The first, third and fourth references can be cited line 44, by indicating :**

*In comparison, on land, systemic approaches are used in studying surface runoff and defining strategies in water resource and flood risk management (Shaikh et al. 2022; Verma et al., 2023; Mehta et al., 2023).*

**The second reference could be mentioned at the very end of the conclusion :**

*As a follow-up of this study, our method may be improved in the future, by exploring the capabilities of artificial intelligence (AI). In addition to large-scale hydrodynamic model outputs and other environmental data, analyses may integrate deep learning method outputs. AI has already provided interesting results in the field of hydrology (Kumar et al., 2023) and for the prediction of storm surges (Tiggeloven et al., 2021).*

**Finally, the references we propose to add are as follows:**

*Hamlington, B. D., Gardner, A. S., Ivins, E., Lenaerts, J. T. M., Reager, J. T., Trossman, D. S., et al. (2020). Understanding of contemporary regional sea-level change and the implications for the future. Reviews of Geophysics, 58, e2019RG000672. https://doi.org/10.1029/2019RG000672*

*Kumar, V., Kedam, N., Sharma, K. V., Mehta, D. J., & Caloiero, T. : Advanced Machine Learning Techniques to Improve Hydrological Prediction: A Comparative Analysis of Streamflow Prediction Models. Water, 15(14), 2572, 2023*

*Mehta, D., Hadvani, J., Kanthariya, D., and Sonawala, P. : Effect of land use land cover change on runoff characteristics using curve number: A GIS and remote sensing approach. International Journal of Hydrology Science and Technology, 16(1), 1-16, 2023.*

*Shaikh, M. M., Lodha, P., Lalwani, P. and Mehta, D.: Climatic projections of Western India using global and regional climate models. Water Practice & Technology, 17(9), 1818-1825, 2022.*

*Tiggeloven, T., Couasnon, A., van Straaten, C. et al. : Exploring deep learning capabilities for surge predictions in coastal areas. Sci Rep 11, 17224. https://doi.org/10.1038/s41598-021-96674-0, 2021.*

*Verma, S., Verma, M. K., Prasad, A. D., Mehta, D., Azamathulla, H. M., Muttil, N., and Rathnayake, U.: Simulating the Hydrological Processes under Multiple Land Use/Land Cover and Climate Change Scenarios in the Mahanadi Reservoir Complex, Chhattisgarh, India. Water, 15(17), 3068, 2023.*

**Anonymous Referee #2**

*1 Overall assessment*

*The paper titled "A systemic and comprehensive assessment of coastal hazard changes: method and application to France and its overseas territories" is an attempt to, as the title suggests, systemic (e.g, coast as a system) and comprehensive assessment of coastal hazard change (in the context of climate change). The paper goes long discussion about its method, then appears to pull observation from some available repositories (in France), pull climate projection from other studies and proposes to provide a systemic (or systematic?) and comprehensive assessment of coastal hazard changes. The final result boils down to a table of regions where qualitative/quantitative measurements from the above-mentioned data are put together, and a mixture of subjective and objective opinion is provided.*

The main contribution of the paper is the proposed method, not the results obtained in the case study. If these results, considered separately or jointly, are also of intrinsic interest, they are mainly presented to demonstrate the interest of our method.

*The objective of this paper is ambitious, and necessary in the context of the risk of multiple coastal hazards, and their unknown/uncertain evolution in the changing climate. I would like to thank the authors to take time to work on this topic. However, unfortunately, after reading the manuscript, I was left with a hollow impression. At the current condition of the manuscript, I do not recommend it's publication in NHESS, and my decision would be to reject.*

**We hope that the adaptations we propose below in response to your comments will help to better highlight the contributions and clarify the objectives of our article.**

Although we have not tried to make a significant contribution to the quantitative evaluation of the physical phenomena, including extreme water levels and their frequencies, we think that our method of evaluating the evolution of the hazard with a more global approach is still of real interest. The concepts of risk and hazard are at the heart of our reflection, and the apprehension of the hazard for complex systems like the Earth are possible, in our opinion, only if systemic approaches complement the analytical approaches (deterministic or probabilistic). We thus feel like this paper is appropriate for a journal that aims to « embrace a holistic Earth system science approach ».

*2 Reasoning for the decision*

*Despite my negative decision, below I have tried to provide a relatively broad reasoning of my decision and some ways the work can be improved (in my opinion). I have also marked some smaller matter in the line-by-line comments. I hope it helps the authors to rethink about their approach, analysis, and presentation.*

Thank you for clarifying the reasons for your suggestion to reject the paper. Below are our answers.

We also thank you for reviewing the document line by line, as these detailed comments will improve the paper on most of the points you report to us.

*The first one is regarding the physics, and the referencing of the existing knowledge of the physical processes that constitute the hazard. There are countless profound claims regarding storm surge characteristics, tide-surge interactions, contribution of wave setup, link between shoreline and hazard and many more are written, but for which very little or often no reference are provided. For each region*

*in France, for practically each component of hazard in question, there are many available literature that should be cited, but it was not the case. Particularly, the consideration of wave is very highlighted in this paper as a novelty. However, the well outdated (and overused) eqn. of 0.2 Hs [1] is too oversimplification of an important factor. One way-around would be to get the best assessment from available studies regarding the scaling, if not there are already existing literature that proposes alternative beach-slope dependent formulations (e.g., [2]). The non-linear interaction between various components - tide, surge, wave-setup are now well established (e.g, [3]) and needs to be well thought too.*

The first reference you cite (Aucan et al., 2018) notes the importance of the effect of waves on the sea level. However, our objective is not to evaluate the evolution of a parameter at a precise time, but to improve the overall assessment of coastal hazards in the long term, taking into account that this assessment is carried out using hypothetical scenarios and that there are high uncertainties related to the multiple factors determining the hazard, which are reinforced by the interactions between these factors. In this context, the precise calculation of the effect of waves on the sea level is of relative importance. In particular, concerning the wave set-up estimate, line 172 stated:

« 0.2 $H_s$ is a generic approximation of the wave set-up (U.S. Army Corps of Engineers, 2002). This equation is a conservative estimate of the wave set-up, which mostly depends on the local slope, breaking wave height, and wave period, and may be closer to 10% of the breaking wave height. Larger values may only be observed at steep sandy beaches (e.g. Martins et al., 2022) or steep shore platforms (Sheremet et al., 2014; Lavaud et al., 2022). This expression is used here since the objective of this work is not to improve existing deterministic methods, but to present a systemic and comprehensive method for assessing the evolution of coastal hazards. »

The method for estimating sea levels is given by default and, if desired, the user of the method can produce other estimates. This position may be justified as follows by completing the introduction of Part 2, with three new paragraphs starting with line 142 (and replacing the last sentence of the paragraph) :

[revised manuscript text omitted]

Please note that advances in modelling are rapid. While our case studies do not take into account the latest modelling developments, if necessary our method can integrate the results provided by the new models.

**On line 173, after equation (4), we will refer to Stockdon et al. (2006) that propose an alternative beach-slope dependent formulation.**

**On line 175, we propose to replace the sentence:**
**« This expression is used here since the objective of this work is not to improve existing deterministic methods, but to present a systemic and comprehensive method for assessing the evolution of coastal hazards. »**
**By:**
**« In the current state of the art, numerical simulations of set-up have been carried out only locally (e.g., Lange et al. (2021), van Ormondt et al. (2021)). Such retrospective simulations are**

computationally expensive (e.g. to simulate accurately set-up, model grid resolution of ~10-50 m may be needed for study sites with large variations in local bathymetry and the wave field). In addition : (i) accurate and high resolution wave set-up modeling requires high resolution bathymetric data (Stephens et al., 2011), which are not available at the spatial scales of our case study; (ii) significant morphological changes are expected in nearshore areas, especially with rising sea levels, which can have a significant effect on the wave set-up at different timescales (e.g. especially for sandy beaches exposed to waves: Ruggiero et al., 2001 ; Thiébot et al., 2012 ; Brivois et al., 2012). Thus, at large spatial (e.g. global) and long temporal (e.g. 21st century) scales, simplified models such as the one used here (Vousdoukas et al., 2018) may provide a first estimate of the expected wave set-up. »

We feel as though the work of Vousdoukas et al. (2018) framework is an appropriate choice for the desired spatial and temporal scales in this study.

*For the overseas areas of France, there are excellent paper that uses sophisticated modelling with thousands of cyclones to quantify hazards (e.g., [4, 5]) - they are needed to be consulted. In broader sense, much more effort must be given to harvest the existing knowledge, particularly over France, the case-study of this paper.*
In principle, since we defend a systemic approach in addition to the analytical approach, our method should promote the use of all the knowledge available in a territory. **We have therefore included additional relevant references in the case studies (these references are presented in our responses to your detailed comments).** You will understand however that our first intention for the presentation of the nine case studies, is to show the form that can take the results (our second intention is to show the differences that may appear in the results obtained on the different sites).

*Secondly, The organisation of the paper is odd. The method section (section 2) is very long, with a lot of reasoning (which reads like a discussion rather than a method), and always referring to things in France - the study area - which is actually presented afterwards (section 3). It appears like, although I hope my guess is wrong, that the paper was first drafted for France, and then it was re-organise to present as a globally applicable method with an application to France. As such, if the Section 3 and Section 2 are switched, the text makes more sense.*

Since the primary objective of the paper is to present a method, section 2 is devoted a detailed presentation of the approach, explaining the underlying concepts (that of risk in particular) and the assessment principles (qualitative systemic approach to accompany the quantitative analytical approach, how to take into account the high uncertainties generated by climate change on the various components of the hazard).
The order and format of presentation of the data associated with the different sites is an integral part of the application of our method. Therefore, it is necessary to maintain the structure of the article "introduction - method - data of study sites and results - discussion - conclusion", even if a plan "introduction - data of study sites - method - results - discussion - conclusion" would have been possible without this constraint.

**In response to your comment (and the comments of the first reviewer), we will present the method more concisely. These changes will reduce the number of references to metropolitan France (only one reference will remain (line 215)) and rebalance with references to other regions of the world (e.g. West Indies (line 222), United States (line 223), La Reunion (line 228), Polar regions (line 253)).**

*Finally, for a paper this ambitious, no data analysis is done, most of the figures are off-the-shelves, and most of the results are table, which are not often compact, with repeated results. I was looking for a map that summarises these coastal hazards over France, but it was disappointing to not find one.*

To give all the necessary follow-up to this comment (and the previous one), we propose to replace the last two paragraphs of the introduction (line 90-99) with the following paragraphs which specify the objectives of the study and its protocol of elaboration:

**« The objective of this paper is to present a comprehensive method to assess the evolution of coastal hazards at regional scales in the context of climate change. The proposed systemic method emphasizes the need to focus on the analysis and interpretation of the modelling results, by putting them into perspective with respect to the biophysical conditions (both current and forecasted).**

**This method was developed as the result of an empirical process, considering diverse situations. This process has the advantage of demonstrating the wide applicability of the proposed method. The case studies used to develop the method are in France and French territories, but they are located in different latitudes (equator, tropics and temperate zones) where they are exposed to different climates and are characterized by different geomorphological configurations (including continental or island). This study highlights that considering the qualitative factors describing the geomorphological context, metocean events and the marine environment in a systemic approach was necessary to assess the evolution of hazards.**

**To ensure that the method can be applied for operational purposes, the use of freely accessible data has been promoted. However, applications of this method should include a thorough bibliographic analyse or even additional observational or modeling studies at the chosen case study sites. It is important to emphasize here that the application of the method depends strongly on the available data, and therefore it is necessary to gather the best data for each site. Since the quantity and quality of data is not the same everywhere, the uncertainties in the results will also vary and must be addressed.**

**Finally, following the application of the method to the case studies, the results obtained using this systemic approach can contribute to improving predictions of the evolution of different types of coastal hazards (shoreline erosion, rapid submersion and/or permanent flooding). »**

**In addition, we can merge and refine some tables to present the data in a more compact format (see our responses to your detailed comments).**

Regarding the representation of the evolution of hazards in France and French territories in the form of a map, this would not bring additional information compared to the text, in particular since regional hazard analyses should extend to the local scale, as indicated in Section 4.

*One approach, that might be of interest for the further development of this paper, would be to do a consensus based assessment, where all the contributing factors listed in this paper are assessed based on existing literature. From there to find how consensus the results are - e.g., IPCC approach. Then, how this consensus differs from the results of the current study - which will potentially identify the research gaps in this line of study.*

Extending the research by the consensus method could be interesting, provided that consensus can be reached, which is not obvious given the diversity of coastal contexts. In all cases, we cannot present in the same paper both our method developed on an empirical basis and other methodological principles developed by consensus. On this point, **we propose to include in the conclusion as a research perspective your suggestion of a consensus evaluation based on bibliographic studies on each of the determining factors.**

Note that two questions you ask in the detailed comments are fundamental for our paper:

-in line 40 : *Why ESL is not sufficient to describe the evolution of the coastal hazards?*

-in line 250-252 : *How does these sentences [relative to the functional collapse of coral reefs] fit to the current discussion of hazard ?*

We plan to respond to this in the introduction of the paper, starting at line 45 (where you correctly pointed out that a transition was missing):

**« To answer this problem [hazard cannot be represented by a single parameter: the maximum water level reached during an event] in the coastal domain, it is necessary to highlight that during a metocean event, the phenomena of flooding and coastal erosion are not only determined by the maximum sea level, but also by coastal waves and currents, and overtopping and overflow discharges over flood protection structures (Formentin et al., 2018; Igigabel et al., 2022). Estimating these discharges and hydrodynamic conditions for the duration of an event requires a good understanding of the physical phenomena that generate the hazard. Retrospective analyses of events help to understand correctly the mechanisms that cause the observed flooding or erosion. For example, by simulating Total Water Levels (TWLs) along the Bight extending from North Carolina to Florida during three historical Tropical Cyclones (TCs) with similar tracks, Hsu et al. (2023) found that the magnitude and duration of the increase in TWLs and wind waves are influenced by TC intensity, translational speed and distance from the shore. In particular, the authors showed that a decrease in TC translation speed led to longer exceedance durations of TWLs, which may result in larger impacts. However, it is not possible to predict deterministically the physical characteristics of future events, nor to assess the corresponding hydrodynamic conditions. To compensate for this, probabilistic approaches have been developed. For example, using a large number of synthetic hurricanes that consider the natural variability in hurricane frequency, size, intensity and track, Krien et al. (2015) estimate the 100-year and 1000-year surge levels for the archipelago of Guadeloupe. Following the same principle, Krien et al. (2017) estimate 100-year surge levels in Martinique for the present climate or considering a potential sea level rise. These results help to determine the necessary levels of protection structures in the short and medium term. However, a single parameter (the maximum water level) is not enough to characterize the hazard and define all the crisis management measures, particularly when water levels exceed the level of protection or when protection structures fail (for example, breaches in levees or dunes). In addition, the accuracy of these estimates decreases in the long term due to: (i) high uncertainties in sea levels beyond 2050 (IPCC, 2019); (ii) the increase in the proportion of high-intensity cyclones worldwide (Masson-Delmotte et al., 2021); and (iii) environmental instability, notably because of geomorphological (e.g., subsidence, coastline retreat) and biological (e.g., degradation of coral reefs and mangroves) changes. These changes modify hydrodynamic and hydrosedimentary processes on the coast both in the long term and during individual events.**

**With the aim of making progress in the global assessment of coastal hazards in the long term, the guiding principle of this paper is to promote the use of the latest advances in research on changes in metocean events and water levels, while also showing the importance of studying other factors whose evolution is more predictable than storm surge and that may be equally important, namely: tidal regimes, geomorphological settings and environmental changes (particularly those modifying hydrodynamic conditions at the coast).**

**Although water levels are not the only parameter to consider, it is necessary to begin by clarifying the definitions of the different levels to which we will refer regularly. »**

**3 Line-by-Line comments**

*1. Abstract: Is appears that "meteocean events" is the main character of your study. Please consider giving a brief definition of what a "metecean event" is in the context of the study.*

**For clarity, by convention, we will use " metocean events" throughout the document to generically name storms, cyclones, and tsunamis (even if the former have no meteorological components).**

*2. L12: Perhaps you meant "metocean" instead of "meteocean"? In the existing literature, I can only find reference to "metocean" which refers to the combined effect of the meteorologic and oceanographic conditions. If "meteocean" (as currently written in the manuscript) was the term you wanted to introduce, please consider introducing it in this line by incorporating briefly the definition to make its meaning clear (compared to "metocean").*

**Idem.**

*3. L27: Please consider adding a few relevant references to the line "Recent research. . . ".*

**We will cite Cazenave and Llovel (2010), Cazenave and Le Cozannet (2013), Hamlington et al. (2020) and Fox-Kemper et al. (2021).**

*4. L36: What does "very likely" refers to in this context? Same as IPCC terminology?*

Yes, the probability is between 90 and 100%.

*5. L40: Why ESL is not sufficient to describe the evolution of the coastal hazards? Please consider brief elaboration of the explanation to come, or provide relevant reference or cases where it was found not enough (e.g., Igigabel et al. 2021 that is cited in L44).*

**Response provided at the end of the general comments.**

*6. L46: Please consider adding a connecting line to indicate for which purpose we need to "First, it is necessary to . . . ".*

**Response provided in the general comments.**

*7. L122: Does "consequence" here means the same as "impacts" as described in IPCC AR5 WG2 report?*

Yes.

*8. L141: "The application of the proposed method . . . " for which purpose? It appears that something is missing from this line.*

**OK, the text has been edited. Please see the response provided in the general comment (Table 1 and the following text).**

*9. L142-144: This statement is interesting and thought-provoking, please elaborate.*

**Response provided at the end of the general comments.**

*10. L184: What would be the 3-maritime facades of France? Perhaps consider adding a bit more somewhere about the coastline of France.*

**OK, the localisation of these facades has been improved through map showing the French territories considered in our study (Figure 2).**

*11. L190: Pickering et al. 2012 - The impact of future sea-level rise on the European Shelf Tides*

**OK, thank you for this reference.**

*12. L201: Is "meteo-oceanic" event is the same as metocean/meteocean event?*

Yes, and **we will prefer « metocean » throughout the paper.**

*13. L213-214: What kind of analysis? How about literature review?*

We will replace the sentence

« First, the analysis of storm surges computed from the national REFMAR database reveals that they are controlled not only by storm tracks but also by the width of the continental shelf and the presence of shallow waters. »

By :

« **Storm surges are controlled not only by the characteristics of metocean events (e.g. for tropical cyclones (TC), the TC intensity, the distance to the TC eye, the TC heading direction and the TC translation (Hsu et al., 2023)), but also by the width of the continental shelf and the relative water depth (Kennedy et al., 2012).** »

**In the next paragraph we will refer to Krien et al. (2015) and Krien et al. (2017) to support the fact that « on the islands of the West Indies, the storm surges rarely exceed 3 m ».**

*14. L229: "maritime facade" is mentioned again here (directly translated from french façade maritime perhaps? it does not seem to exist in English), please elaborate what it means somewhere in the text.*

« Facade » is a word also used by English native speakers. **The localisation of these facades on a map will definitely help to understand the text.** Thank you.

*15. L231: Please consider providing proper journal/article reference (which exists) instead of a generic website from noaa.*

**We will refer to Flather (2001), Rego and Li (2010) and Kennedy et al. (2012).**

*16. L234: Please consider adding a real example from published literature.*

**We will refer to Bertin et al. (2012), Krien et al. (2015) and Krien et al. (2017).**

*17. L247: "should" -> "expected to".*

To avoid redundancy with « expected », we propose « may ».

*18. L250-252: How does these sentences fit to the current discussion of hazard? Please consider rewriting/revising/deleting.*

As indicated in the answers to general comments, biological changes should be taken into account in the assessement of hazard changes.

**In complement, we could add in the discussion (L. 581), as indicated in Krien et al. (2017) : « Moreover, coastal ecosystems such as mangroves, coral reefs or seagrass beds may not be able to adapt to climate change (e.g., Waycott et al., 2009; Wong et al., 2014), which could have large impacts on coastal hazards (e.g., Alongi, 2008; Wong et al., 2014). »**

*19. L280: Why French coast was chosen to demonstrate the method?*

Response provided in the general comments.

*20. Figure 2: Please consider a bit more elaborate caption and add reference to the figure if it is adapted from somewhere else.*

OK.

*21. Figure 2: Please consider putting different colour for so called "marine facade".*

OK, good idea. Thank you. **Finally, we propose to use lines of dots, dashes and crosses to represent the 3 facades, so that this representation can be readable on a black and white print.**

*22. L280-300: I do not understand the objective of the paragraph regarding the GMSL projections. Neither in your equation of ESL (eq 3.) nor in the list presented in L270 there is GMSL present.*

OK, you're right. In an earlier version of the paper, this text was positioned at line 144.

**Here, we propose « The choice of the scenario and the time scales should take into account the existence of the high value assets at the considered coasts (coastal cities, port and industrial facilities) and the strong uncertainties about the contributions of ice caps to the rise in water levels (Bamber et al., 2019; Dayan et al., 2021). For these two reasons, we use the SSP5-8.5 scenario. »**

**And the rest of the paragraph will be moved to section 2 to explain the high uncertainties in sea level changes.**

*23. L304: Consider giving a 1-2 line summary of Vousdoukas et al. (2018) framework.*

We propose to replace:

« Projections of waves and storm surges were based on hydrodynamic simulations driven by atmospheric forcing from six Coupled Model Intercomparison Project Phase 5 (CMIP5) climate models. »

By:

**« The framework developed by Vousdoukas et al. (2018) is used to evaluate the contributions of each of these components, of which the baseline values are calculated in global reanalyses of waves and storm surges. Then, CMIP5 models are used to estimate future relative changes to the meteorological water levels. Lastly, changes in sea level, the astronomical tide, and meteorological water levels are combined to produce the ESL values. »**

*24. L305: Are these projections of waves and storm surges published already? Has there been any bias correction done to CMIP5 data?*

No bias correction is needed since we are using the CMIP5 simulation results only to obtain relative changes. In addition the results have been validated in the following articles:

Vousdoukas, M.I., Mentaschi, L., Voukouvalas, E., Verlaan, M., Jevrejeva, S., Jackson L. P. & Feyen, L., 2018. Global probabilistic projections of extreme sea levels show intensification of coastal flood hazard. Nat Commun 9, 2360. https://doi.org/10.1038/s41467-018-04692-w.

Mentaschi, L., M. I. Vousdoukas, E. Voukouvalas, A. Dosio, and L. Feyen, 2017. Global changes of extreme coastal wave energy fluxes triggered by intensified teleconnection patterns, Geophys. Res. Lett., 44, 2416–2426, doi:10.1002/2016GL072488.

Vousdoukas, M. I., Mentaschi, L., Voukouvalas, E., Verlaan, M., and Feyen, L., 2017. Extreme sea levels on the rise along Europe's coasts. Earth's Future, 5, 304–323. doi:10.1002/2016EF000505.

*25. L308-310: Where are these projection coming from? I do not see a reference here. Is it from CMIP 6 project?*

As indicated, it is CMIP5 and not CMIP6.

*26. Table 2: Please add lon,lat location of the tide gauges. Please also add another column with tidal range.*

**Good idea to add latitude and longitude.**

Tidal ranges are presented in 3.1.2. To keep the structure of the section and avoid redundancy, we prefer not to add this information in Table 2.

| | Latitude | Longitude | Projection of RSL rise (m) | |
| --- | --- | --- | --- | --- |
| | | | 2050 | 2100 |
| Calais | 50.972122801049395 | 1.8400588679271384 | 0.19 | 0.86 |
| Le Havre | 49.485774012966544 | 0.0897840202510471 | 0.19 | 0.87 |
| Saint-Malo | 48.641170797005124 | -2.0313888026241402 | 0.19 | 0.87 |
| Brest | 48.368792381060274 | -4.4887286031866935 | 0.19 | 0.83 |
| La Rochelle | 46.14879125637811 | -1.1691588857635562 | 0.16 | 0.76 |
| Saint-Jean-de-Luz | 43.39842588942395 | -1.676715829943722 | 0.17 | 0.79 |
| Port Vendres | 42.52411089711506, | 3.1143835886292806 | 0.16 | 0.76 |
| Sète | 43.39330679508365 | 3.699492031443802 | 0.16 | 0.76 |
| Marseille | 43.29569227471328 | 5.352448215467127 | 0.17 | 0.78 |
| Saint-Pierre (Saint-Pierre-et-Miquelon) | 46.786272435631275 | -56.16190646722868 | 0.19 | 0.83 |
| Pointe-à-Pitre (Guadeloupe) | 16.23300045952869 | -61.53571250198115 | 0.20 | 0.90 |
| Cayenne (French Guiana) | 4.93572687841612 | -52.340676954198194 | 0.20 | 0.90 |

| | | | | |
|---|---|---|---|---|
| Pointe des Galets (La Réunion) | -20.936178918145654 | 55.280686667086655 | 0.19 | 0.92 |
| Papeete (French Polynesia) | -17.53479287238126 | -149.58674796473545 | 0.20 | 0.91 |

*27. L322: In the "standard classification" are there more than 3 types? Please add a reference to standard classification, like Book of Pugh and Woodworth 2014.*

**We will refer to Masselink and Short (1993).**

*28. Figure 3: The figure contains an incomplete description. The Mediterranean is missing, so is the other french islands. What is type of the data? How it was generated? Model? Altimetry? Tide gauges? Please provide further detail.*

The objective of our article is not to detail as much as possible the quantitative assessment of the different components of the water level. As for the wave set-up, the goal is to provide the order of magnitude.

The map is provided by the SHOM, which produces the reference maritime and coastal geographic information in France. It aims at illustrating tidal ranges along macrotidal coastlines. That's why the information is limited to English Channel and the Atlantic. The other coasts are microtidal or mesotidal.

*29. Table 3: Please consider combining Table 3 with Table 2.*

To keep the structure of the section, we would prefer not to combine Table 3 with Table 2.

*30. L359: Please provide the link for ReefTEMPS and DYNALIT services.*

**DYNALIT: https://www.dynalit.fr/ and REEFTEMPS: https://www.reeftemps.science/**

*31. L361: Why infragravity waves with Hs ~1m can be superimposed on this set-up? Reference?*

Bertin et al. (2020) showed that infragravity waves of Hs~1.5 m were superimposed to a surge (wind + wave setup) of 0.5 to 1.0 m.

Baumann et al. (2017) showed that IG waves could reach 2 m during storm Hercules.

In Truc Vert, Ruessink (2010) measured IG waves of Hs~1.5 m during storm Johanna, he did not provide estimates of wave setup but Nicolae-Lerma et al. (2017) provided estimates of wave setup up to 0.8 m for this storm.

**To keep the text short, we propose to add only the reference to Bertin et al. (2020).**

*32. L363: "This information" -> which information? Which geo-morphological configuration?*

**« This information confirms » can be replaced by « These observations confirm ».**

*33. Table 4-5: Are the results taken directly from Vousdoukas et al. (2018) or reanalyzed? It is not clear to me.*

As indicated in lines 304, 366 and 386, the results are taken directly from the framework developped by Vousdoukas et al. (2018). The reasons for this choice are explained in the general comments and will be transcribed in the paper.

*34. L380: Reference for this claim about Mediterranean? Or is it analyzed somewhere in this manuscript? It is not clear.*

**We will refer to Vitousek et al. (2017), since this reference is cited concerning this subject in the introduction.**

*35. Why Table 5 and Table 7 is separated?*

The idea is to analyse separately mainland France and overseas territories. If you want, we can merge the tables and adapt the texts accordingly.

*36. Same question as above for Table 4 and Table 6.*

Idem.

*37. L405: Repeated, not needed.*

**OK.**

*38. Table 8: Please add relevant reference to another column. Since it is a "qualitative" assessment, without reference it does not hold enough validity. Adding reference to each cases will also add values to all the past regional studies that are done over these various regions. Same goes for related text, where there appears to be no references currently (L428-446). In addition, it is not clear how these subjective labels are provided - e.g., Very high, High etc.*

**The diversity of situations does not really make it possible to set up a calibration and weighting system. Here our method aims to help experts to make a structured judgment on the « surge potential associated with the geomorphological configuration » based on a list of criteria. We agree that experts should also be encouraged to integrate past study results in their jugments. To this end past regional studies should indeed appear in Table 8. Here are the references we propose to cite on the different coastlines considered for our case studies.**

**For the three metropolitan French facades :**

[revised manuscript text omitted]

*39. Section 3.2.3: How these factors are taken into account? Any subjective or objective comparison?*

In the proposed method, we suggest checking that such factors (changes in the swell climate related to the decrease in seasonal sea ice extent, and coral reefs degradation) are not likely to significantly change the hydraulic conditions in the long term. It is difficult to make comparisons with the influence of other factors (e.g. geomorphological configuration, reference metocean event, tidal range, frequency and intensity of extreme climate events).

*40. Why Table 9 and 10 are separated? It seems the tide gauge stations are now aggregated. Why it is so?*

The goal is to avoid too much information in the same table.

The aggregated data provide regional conclusions. If conclusions are sought at a more local scale, as explained in Section 4, it is necessary to extend the investigations.

*41. It is not very clear how in Table 9 and 10, the "important" and "most important" labels are applied. Are they coming from assessment of available literature?*

Indeed, it would be better to qualify the factors by the terms "slightly detrimental ", " detrimental " and "highly detrimental".

**In this method, each factor should not be considered independently. On the contrary, the joint effects of the different factors should be assessed to understand the dynamics of the system. However, the diversity of the different observed contexts prevents proposing a quantitative calibration and weighting system. This is the main reason why this method is called « systemic », in reference to the study of a system (the coast). However, this method should not be called « systematic ». Even though the analysis framework incorporates multiple factors, the expert in charge of the study of a particular site should consider the qualitative and quantitative factors together. In summary, this method aims to help experts to make a structured evaluation.**

**We propose to insert this explanation at the end of the presentation of the method (line 279).**

42. L614: I believe GMSL is not taken into account here as global sense, rather it was included into RSL. Is it?

**You're right.**

*43. L665: How the impact of sea level changes on human communities are evaluated in paper? I do not see it. I do not also see where the "anthropogenic structures" are considered, and how it was considered.*

**Anthropogenic structures are mentioned explicitly in section 4.1 for estuaries and polar regions. We can also mention them for sandy coasts.**

**Additional references**

Alongi, D. M.: Mangrove forests: resilience, protection from tsunamis, and response to global climate change, Estuar. Coast. Shelf S., 76, 1–13, 2008.

Arns A., Wahl T., Dangendorf S., Jensen J.: The impact of sea level rise on storm surge water levels in the northern part of the German Bight. Coast Eng 96:118–131. https://doi.org/10.1016/j.coastaleng.2014.12.002, 2015

Arns, A., Wahl, T., Wolff, C., Vafeidis A., Haigh I., Woodworth P., Niehüser S. and Jensen J.: Non-linear interaction modulates global extreme sea levels, coastal flood exposure, and impacts. Nat Commun 11, 1918. https://doi.org/10.1038/s41467-020-15752-5, 2020.

Aucan J., Hoeke R. K., Storlazzi C. D., Stopa J., Wandres M., and Lowe R.: Waves do not contribute to global sea-level rise. Nature Climate Change, 9(1):2–2, December 2018.

Bertin X., Martins K., de Bakker A., Chataigner T., Guérin T., et al.: Energy Transfers and Reflection of Infragravity Waves at a Dissipative Beach Under Storm Waves. Journal of Geophysical Research. Oceans, 125 (5), pp.e2019JC015714, 2020.

Brivois O., Idier D., Thiébot J., Castelle B., Le Cozannet G., Calvete D.: On the use of linear stability model to characterize the morphological behaviour of a double bar system. Application to Truc Vert beach (France). CR Geosci 344:277–287. https ://doi.org/10.1016/j.crte.2012.02.004, 2012.

Cazenave, A. and Llovel, W.: Contemporary sea level rise. Ann. Rev. Marine Sci. 2, 145‑173, 2010.

Cazenave, A. and Le Cozannet, G.: Sea level rise and its coastal impacts, Earth's Future, 2, 15–34. doi:10.1002/2013EF000188, 2013.

Flather, R.A.: Storm surges. In: Steele, J., Thorpe, S., Turekian, K. (Eds.), En- cyclopedia of Ocean Sciences. Academic, San Diego, California, pp.2882–2892, 2001.

Formentin, S.M. and Zanuttigh, B.: A new method to estimate the overtopping and overflow discharge at over-washed and breached dikes. Coast. Eng., 140, 240–256. https://doi.org/10.1016/j.coastaleng.2018.08.002, 2018.

Fox-Kemper, B. et al., 2021. Ocean, cryosphere and sea level change. In IPCC AR6 (eds Fox-Kemper, B. et al.) 1211–1362. Cambridge University Press.

[revised manuscript text omitted]

**References cited in this document that will not be included in the paper**

Baumann J., Chaumillon E., Bertin X., Schneider J.-L., Guillot B., Schmutz M. : Importance of infragravity waves for the generation of washover deposits, Marine Geology, Volume 391, Pages 20-35, ISSN 0025-3227, https://doi.org/10.1016/j.margeo.2017.07.013, 2017.

Nicolae-Lerma A., Pedreros R., Robinet A., Sénéchal N.: Simulating wave setup and runup during storm conditions on a complex barred beach, Coastal Engineering, Volume 123, Pages 29-41, ISSN 0378-3839, https://doi.org/10.1016/j.coastaleng.2017.01.011, 2017.

Ruessink, B. G.: Observations of turbulence within a natural surf zone. Journal of Physical Oceanography 40, 2696–2712, 2010.